# Quantum speedups for stochastic optimization

Aaron Sidford and Chenyi Zhang

{sidford,chenyiz}@stanford.edu

## Abstract

We consider the problem of minimizing a continuous function given given access to a natural quantum generalization of a stochastic gradient oracle. We provide two new methods for the special case of minimizing a Lipschitz convex function. Each method obtains a dimension versus accuracy trade-off which is provably unachievable classically and we prove that one method is asymptotically optimal in low-dimensional settings. Additionally, we provide quantum algorithms for computing a critical point of a smooth non-convex function at rates not known to be achievable classically. To obtain these results we build upon the quantum multivariate mean estimation result of Cornelissen et al. [25] and provide a general quantum variance reduction technique of independent interest.

## 1 Introduction

Stochastic optimization is central to modern machine learning. Stochastic gradient descent (SGD), and its many variants, are used broadly for solving challenges in data science and learning theory. In theory, SGD and stochastic optimization, have been the subject of decades of extensive study. [19, 68, 27] established that SGD achieves optimal rates for minimizing Lipschitz convex functions[1] (even in one dimension) and stochastic optimization methods have been established for a range of problems [72, 70, 73, 74]. More recently, the complexity of stochastic gradient methods for smooth non-convex optimization, e.g., critical point computation, were established [3, 29, 30, 49, 50].

Given the foundational nature of stochastic optimization and the potential promise and increased study of quantum algorithms, it is natural to ask whether quantum computation could enable improved rates for solving these problems. There has been work studying whether access to quantum counterparts of classic optimization oracle can yield faster rates for semidefinite programs [13, 12, 43, 75, 76, 54], convex optimization [21, 5, 20], and non-convex optimization [23, 63, 80]. Notably, [52] showed that with just access to the quantum analog of an *zeroth-order oracle*, i.e., an oracle that when queried at a point outputs the value of the function at that point, it is possible to simulate access to a classic gradient oracle with a single query. This tool immediately yields improved rates for, e.g., convex function minimization, with a zeroth-order oracle.

Unfortunately, despite this progress and the established power of quantum evaluation oracles, obtaining further improvements has been challenging. A line of work established a variety of striking lower bounds ruling out quantum speedups for fundamental optimization problems [31, 32, 82]. For example, [31] showed that when given access to the quantum analog of a *first-order oracle*, i.e., an oracle that when queried at a point outputs the value of the function as well as the gradient at that point, quantum algorithms have no improved rates for non-smooth convex optimization over GD and SGD when the dimension is large. [82] extended this result to the non-convex setting and

---

[1]We assume all objective functions are differentiable. Similar to related work, see e.g., [15], our results can be generalized to non-differentiable settings since convex functions are almost everywhere differentiable and our algorithms and corresponding convergence analysis are robust to polynomially small numerical errors.

37th Conference on Neural Information Processing Systems (NeurIPS 2023).

showed that given access to the quantum analog of a stochastic gradient oracle, quantum algorithms have no improved rates for finding critical points over SGD when the dimension is large.

In spite of these negative results, we nevertheless ask, *for stochastic optimization are quantum-speedups obtainable?* Our main result is an answer to this question in the affirmative for dimension-dependent algorithms. We provide two different quantum algorithms for stochastic convex optimization (SCO) which provably outperform optimal classic algorithms. Furthermore, we provide a quantum algorithm for computing the critical point of a smooth non-convex function which improves upon the state-of-the-art. We obtain these results through a new general quantum-variance reduction technique built upon the quantum multivariate mean estimation result of Cornelissen et al. [25] and the multilevel Monte Carlo (MLMC) technique [41, 10, 7]. We complement these results with lower bounds showing that one of them is asymptotically optimal in low-dimensional settings.

**General notation.** We use $\| \cdot \|$ to denote the Euclidean norm and let $\mathbb{B}_R(\mathbf{x}) := \{\mathbf{y} \in \mathbb{R}^d \colon \|\mathbf{y} - \mathbf{x}\| \leq R\}$ and $[T] := \{1, \ldots, T\}$. We use bold letters, e.g., $\mathbf{x}, \mathbf{y}$, to denote vectors and capital letters, e.g., $A, B$, to denote matrices. For a $d$-dimensional random variable $X$, we refer to the trace of the covariance matrix of $X$ as its variance, denoted by $\mathrm{Var}[X]$. For $f \colon \mathbb{R}^d \to \mathbb{R}$, we let $f^* := \inf_{\mathbf{x}} f(\mathbf{x})$ and call $\mathbf{x} \in \mathbb{R}^d$ $\epsilon$-*(sub)optimal* if $f(\mathbf{x}) \leq f^* + \epsilon$ and $\epsilon$-*critical* if $\|\nabla f(\mathbf{x})\| \leq \epsilon$. Moreover, we call a random point $\mathbf{x} \in \mathbb{R}^d$ *expected* $\epsilon$-*(sub)optimal* if $\mathbb{E} f(\mathbf{x}) \leq f^* + \epsilon$ and *expected* $\epsilon$-*critical* if $\mathbb{E}\|\nabla f(\mathbf{x})\| \leq \epsilon$. $f \colon \mathbb{R}^d \to \mathbb{R}$ is *L-Lipschitz* if $f(\mathbf{x}) - f(\mathbf{y}) \leq L\|\mathbf{x} - \mathbf{y}\|$ for all $\mathbf{x}, \mathbf{y} \in \mathbb{R}^d$ and $\ell$-*smooth* if $\|\nabla f(\mathbf{x}) - \nabla f(\mathbf{y})\| \leq \ell\|\mathbf{x} - \mathbf{y}\|$ for all $\mathbf{x}, \mathbf{y} \in \mathbb{R}^d$. For two matrices $A, B \in \mathbb{R}^{d \times d}$, $A \preceq B$ denotes that $\mathbf{x}^T A \mathbf{x} \leq \mathbf{x}^T B \mathbf{x}$ for all $\mathbf{x} \in \mathbb{R}^d$. We use $\widetilde{\mathcal{O}}$ to denote the big-$\mathcal{O}$ notation omitting poly-logarithmic factors in $\epsilon, \epsilon', d, \sigma, \hat{\sigma}, R$, and $L$. When applicable, we use $|\mathrm{garbage}(\cdot)\rangle$ to denote possible garbage states[2] that arise during the implementation of a quantum oracle (e.g., in Definitions (1) and (2)).

## 1.1 Quantum stochastic optimization oracles

Here we formally define quantum stochastic optimization oracles that we study in this work.

**Qubit notation.** We use $|\cdot\rangle$ to represent input or output registers made of qubits that could be in *superpositions*. In particular, given $m$ points $\mathbf{x}_1, \ldots, \mathbf{x}_m \in \mathbb{R}^d$ and a coefficient vector $\mathbf{c} \in \mathbb{C}^m$ with $\sum_{i \in [m]} |c_i|^2 = 1$, the quantum register could be in the quantum state $|\psi\rangle = \sum_{i \in [m]} c_i |\mathbf{x}_i\rangle$, which is superposition over all these $m$ points at the same time. If we measure this state, we will get each $\mathbf{x}_i$ with probability $|c_i|^2$. Furthermore, to model a classical probability distribution $p$ over $\mathbb{R}^d$ quantumly, we can prepare the quantum state $\int_{\mathbf{x} \in \mathbb{R}^d} \sqrt{p(\mathbf{x}) \mathrm{d}\mathbf{x}} |\mathbf{x}\rangle$. If we measure this state, the measurement outcome would follow the probability density function $p$.

**Quantum random variable access.** We say that we have *quantum access to a $d$-dimensional random variable $X$* if we can query the following *quantum sampling oracle* of $X$ that returns a quantum superposition over the probability distribution of $X$ defined as follows.[3]

**Definition 1** (Quantum sampling oracle)**.** *For a $d$-dimensional random variable $X$, its quantum sampling oracle $O_X$ is defined as*

$$O_X |0\rangle \to \int_{\mathbf{x} \in \mathbb{R}^d} \sqrt{p_X(\mathbf{x}) \mathrm{d}\mathbf{x}} |\mathbf{x}\rangle \otimes |\mathrm{garbage}(\mathbf{x})\rangle, \tag{1}$$

*where $p_X(\cdot)$ represents the probability density function of $X$.*

The garbage state in Definition 1 is a quantum analogue of classical garbage information that arises when preparing the classical sampling oracle of $X$. When implementing the quantum sampling

---

[2]The garbage state is a quantum analogue of classical garbage information that arises when preparing the classical stochastic gradient oracle which cannot be erased or uncomputed in general. In this work, we consider a general model where we make no assumption on the garbage state. See e.g., [42] for a similar discussion of this standard use of garbage quantum states.

[3]Throughout this paper, whenever we have access to a quantum oracle $O$, we assume that it is a unitary operation and that we also have access to its corresponding inverse operation, denoted as $O^{-1}$, that satisfies $O^{-1}O = OO^{-1} = I$. These are a standard assumption, either explicitly or implicitly, in prior work on quantum algorithms, see e.g., [14, 44, 25].

oracle in quantum superpositions however, this garbage information will appear in a quantum state and cannot be erased or uncomputed in general. In this work, we consider a general model where we make no assumption on the garbage state. See e.g., [42] for a similar discussion of this standard use of garbage quantum states.

Observe that if we directly measure the output of $O_X$, it will collapse to a classical sampling access to $X$ that returns random vectors with respect to the probability distribution $p_X$.

**Quantum stochastic gradient oracle.** When considering the problem of optimizing a function $f \colon \mathbb{R}^d \to \mathbb{R}$, we are often given access to a stochastic gradient oracle that returns a random vector from the probability distribution of the stochastic gradient.

**Definition 2** (Stochastic gradient oracle (SGO)). *For $f \colon \mathbb{R}^d \to \mathbb{R}$, its* stochastic gradient oracle (SGO) $\mathcal{C}_\mathbf{g}$ *is defined as a random function that when queried at* $\mathbf{x}$*, samples a vector* $\mathbf{g}(\mathbf{x})$ *from a probability distribution* $p_{f,\mathbf{x}}(\cdot)$ *over* $\mathbb{R}^d$ *that satisfies*

$$\mathbb{E}_{\mathbf{g}(\mathbf{x}) \sim p_{f,\mathbf{x}}} \mathbf{g}(\mathbf{x}) = \nabla f(\mathbf{x}), \quad \forall \mathbf{x} \in \mathbb{R}^d.$$

*We say the oracle is $L$-bounded if*

$$\mathbb{E}_{\mathbf{g}(\mathbf{x}) \sim p_{f,\mathbf{x}}} \|\mathbf{g}(\mathbf{x})\|^2 \le L^2, \quad \forall \mathbf{x} \in \mathbb{R}^d,$$

*and we say the oracle has* variance $\sigma^2$ *if*

$$\mathbb{E}_{\mathbf{g}(\mathbf{x}) \sim p_{f,\mathbf{x}}} \|\mathbf{g}(\mathbf{x}) - \nabla f(\mathbf{x})\|^2 \le \sigma^2, \quad \forall \mathbf{x} \in \mathbb{R}^d.$$

In this paper, we further assume quantum access to a stochastic gradient oracle, or access to a *quantum stochastic gradient oracle* for brevity, that upon query returns a quantum superposition over the probability distribution $p_{f,\mathbf{x}}(\mathbf{v})$.

**Definition 3** (Quantum stochastic gradient oracle (QSGO)). *For $f \colon \mathbb{R}^d \to \mathbb{R}$, its quantum stochastic gradient oracle (QSGO) is defined as*

$$O_\mathbf{g} \ket{\mathbf{x}} \otimes \ket{0} \to \ket{\mathbf{x}} \otimes \int_{\mathbf{v} \in \mathbb{R}^d} \sqrt{p_{f,\mathbf{x}}(\mathbf{v})\mathrm{d}\mathbf{v}} \ket{\mathbf{v}} \otimes \ket{\mathrm{garbage}(\mathbf{v})}, \tag{2}$$

*where $p_{f,\mathbf{x}}(\cdot)$ is as defined in Definition 2.*

Observe that if we directly measure the output of $O_\mathbf{g}$, it will collapse to a classical stochastic gradient oracle that randomly returns a stochastic gradient at $\mathbf{x}$.

Another standard assumption in previous works [1, 29, 30, 50] that the stochastic gradient can be queried *simultaneously*, which means that the algorithm can choose the random seed $\omega$ that is being queried. We assume that in such setting there is an explicit probability distribution $\omega$ such that $\mathbb{E}_\omega[\mathbf{g}(\mathbf{x}, \omega)] = \nabla f(\mathbf{x})$. Similarly, we define the quantum access to stochastic gradients allowing simultaneous queries, or access to a *quantum stochastic gradient oracle with simultaneous queries* for brevity, that upon query returns $\mathbf{g}(\mathbf{x}, \omega)$ in a quantum state.

**Definition 4** ($\sigma$-SQ-QSGO). *For $f \colon \mathbb{R}^d \to \mathbb{R}$ with its stochastic gradient $\mathbf{g}(\mathbf{x}, \omega)$ indexed by random seed $\omega$ that satisfies*

$$\mathbb{E}_\omega \mathbf{g}(\mathbf{x}, \omega) = \nabla f(\mathbf{x}) \quad, \quad \mathbb{E}_\omega \|\mathbf{g}(\mathbf{x}, \omega) - \nabla f(\mathbf{x})\|^2 \le \sigma^2, \quad \forall \mathbf{x} \in \mathbb{R}^d, \quad and$$

$$\mathbb{E}_\omega \|\mathbf{g}(\mathbf{x}, \omega) - \mathbf{g}(\mathbf{y}, \omega)\|^2 \le \ell^2 \|\mathbf{x} - \mathbf{y}\|^2, \quad \forall \mathbf{x}, \mathbf{y} \in \mathbb{R}^d \tag{3}$$

*its simultaneously queriable, $\sigma$-mean-squared smooth quantum stochastic gradient oracle ($\sigma$-SQ-QSGO) is defined as*

$$O_\mathbf{g}^S \ket{\mathbf{x}} \otimes \ket{\omega} \otimes \ket{0} \to \ket{\mathbf{x}} \otimes \ket{\omega} \otimes \ket{\mathbf{g}(\mathbf{x}, \omega)} \otimes \ket{\mathrm{garbage}(\mathbf{x}, \omega)}. \tag{4}$$

## 1.2 Results

Here we present our main results on new quantum algorithms for stochastic optimization. Our results and the prior state-of-the-art are summarized in Table 1. Further, we discuss new quantum lower bounds that we establish for quantum variance reduction and stochastic convex optimization. Our algorithmic results leverage a common technique for quantum variance reduction introduced in the next Section 2. This technique uses a combination of the quantum multivariate mean estimation result of Cornelissen et al. [25] and multilevel Monte Carlo (MLMC) [41, 10, 7].

| Setting | Queries | Output | Method |
|---|---|---|---|
| Convex | $\epsilon^{-2}$ | $\epsilon$-optimal | SGD |
| Convex | $d^{5/8}\epsilon^{-3/2}$ | $\epsilon$-optimal | **Our Result** (Q-AC-SA, Algorithm 2) |
| Convex | $d^{3/2}\epsilon^{-1}$ | $\epsilon$-optimal | **Our Result** (Q-SCP, Section 4) |
| Non-convex (Bounded variance) | $\epsilon^{-4}$ | $\epsilon$-critical | Randomized SGD [40] |
| Non-convex (Bounded variance) | $d^{1/2}\epsilon^{-3}$ | $\epsilon$-critical | **Our Result** (Q-SGD, Algorithm 6) |
| Non-convex (Mean-squared smoothness) | $\epsilon^{-3}$ | $\epsilon$-critical | SPIDER [29] |
| Non-convex (Mean-squared smoothness) | $d^{1/2}\epsilon^{-5/2}$ | $\epsilon$-critical | **Our Result** (Q-SPIDER, Algorithm 7) |

Table 1: Comparison between our quantum algorithms and state-of-the-art classical algorithms for Problem 1 (the convex setting) and Problem 2 (the non-convex setting) where, for simplicity, we assume $R = L = 1$ for Problem 1 and $\ell = \sigma = \delta = 1$ for Problem 2. The $\widetilde{\mathcal{O}}$ symbol was omitted in the "Queries" column.

**Stochastic convex optimization.** In this work we consider the quantum analog of the standard stochastic convex optimization (SCO) problem defined as follows.

**Problem 1** (Quantum stochastic convex optimization (QSCO)). *In the quantum stochastic convex optimization (QSCO) problem we are given query access to an L-bounded QSGO $O_{\mathbf{g}}$ (see Definition 3) for a convex function $f \colon \mathbb{R}^d \to \mathbb{R}$ whose minimum is achieved at $\mathbf{x}^*$ with $\|\mathbf{x}^*\| \leq R$ and must output an expected $\epsilon$-optimal point.*

Classically, it is known that simple stochastic gradient descent, e.g. $x_{t+1} = x_t - \eta g_t$, can solve QSCO with $\mathcal{O}(\epsilon^{-2})$ queries. Further, this bound is known to be optimal in the worst case [69, 77].

Nevertheless, we develop two quantum algorithms for Problem 1 in Section 3 and Section 4, respectively. The query complexities of these algorithms are summarized in the following Theorem 1. In comparison to the optimal classical query complexity of $\mathcal{O}(\epsilon^{-2})$ queries, our algorithms obtain an improved dependence in terms of $\epsilon$ at the cost of a worse dependence on the dimension, $d$; Theorem 1 shows that a quadratic speedup is achievable when $d$ is constant.

**Theorem 1** (Informal version of Theorem 5 and Corollary 1). *Problem 1 can be solved using an expected $\widetilde{\mathcal{O}}(\min\{d^{5/8}(LR/\epsilon)^{3/2}, d^{3/2}LR/\epsilon\})$ queries.*

We complement Theorem 1 with the following lower bound on the query complexity for QSCO.

**Theorem 2** (Informal version of Theorem 8). *For any $\epsilon \leq \mathcal{O}(d^{-1/2})$, any quantum algorithm that solve Problem 1 with probability at least $2/3$ makes at least $\Omega(d^{1/2}\epsilon^{-1})$ queries in the worst case. For any $\Omega(d^{-1/2}) \leq \epsilon \leq 1$, any quantum algorithm that solves Problem 1 with success probability at least $2/3$ must make at least $\Omega(\epsilon^{-2})$ queries in the worst case.*

Theorem 2 shows that the $\widetilde{\mathcal{O}}(d^{3/2}LR/\epsilon)$ rate that we obtain is asymptotically optimal for $d = \widetilde{\mathcal{O}}(1)$. A key open problem is whether the dimension dependence in either our upper bounds (Theorem 5 and Corollary 1) or our lower bound (Theorem 8) can be improved.

**Stochastic critical point computation.** We also develop quantum algorithms for finding critical points, i.e., points with small gradients, of (possibly) non-convex functions.

**Problem 2** (Quantum stochastic critical point computation (QSCP)). *In the quantum stochastic critical point computation (QSCP) problem, for an $\ell$-smooth (posssibly) non-convex $f \colon \mathbb{R}^d \to \mathbb{R}$ satisfying $f(\mathbf{0}) - \inf_{\mathbf{x}} f(\mathbf{x}) \leq \Delta$ we are given query access to one of the following two oracles*

1. *(Bounded variance setting). A QSGO $O_{\mathbf{g}}$ with variance $\sigma^2$ (see Definition 3), or*

2. *(Mean-squared smoothness setting). A $\sigma$-SQ-QSGO $O_{\mathbf{g}}^S$ (see Definition 4),*

*and must output an expected $\epsilon$-critical point.*

Leveraging [40] and [29] we develop two quantum algorithms that solve Problem 2 in the bounded variance setting and the mean-squared smoothness setting, and obtain the following result.

**Theorem 3** (Informal version of Theorem 6 and Theorem 7). *In the bounded variance setting, Problem 2 can be solved using an expected $\widetilde{\mathcal{O}}\left(\Delta\ell\sigma d^{1/2}\epsilon^{-3}\right)$ queries. In the mean-squared smoothness setting, Problem 2 can be solved using an expected $\widetilde{\mathcal{O}}\left(\ell\Delta(d\sigma)^{1/2}\epsilon^{-5/2}\right)$ queries.*

In the bounded variance setting, Problem 2 can be solved using $\mathcal{O}(\epsilon^{-4})$ queries to a classical SGO with variance $\sigma^2$ (Definition 2) [40], which is known to be optimal [6]. In comparison, our algorithm improves in terms of $\epsilon$ and achieves a quantum speedup when $d \leq \mathcal{O}(\epsilon^{-2})$. In the mean-squared smoothness setting, Problem 2 can be solved using $\mathcal{O}(\epsilon^{-3})$ queries to a classical stochastic gradient oracle with variance $\sigma^2$ that satisfies if it satisfies (3) (Definition 2) [29], which is known to be optimal [6]. In comparison, our algorithm improves in terms of $\epsilon$ and achieves a quantum speedup when $d \leq \mathcal{O}(\epsilon^{-1})$.

**Quantum zeroth-order oracles.**   Throughout the paper, we focus on stochastic gradient oracles in correspondence with classical work on stochastic optimization. However, it is worth noting that in certain cases our results extend gracefully to quantum stochastic zeroth order oracles. For example, when the objective function exhibits a finite-sum structure and we have access each component function individually through a quantum zeroth-order oracle, we can achieve an SQ-QSGO (Definition 4) with just a single query, utilizing quantum gradient estimation [52]. However, in other cases, the correspondence is less clear. For instance, if we are given a quantum stochastic zeroth-order oracle where the function value is obfuscated by some external noise, quantum gradient estimation [52] is not directly applicable. Further study could be an interesting direction for future work.

**Practicality.**   Regarding the utility of our algorithm in practical situations, note that our quantum oracles in Definition 3 and Definition 4 are defined as direct, natural generalizations of the corresponding classical oracles. Considering such quantum generalizations of classical oracles is standard in the literature, see e.g., [21, 81]. There are standard techniques for implementing such quantum analogs of classical oracles (in theory for now given the current state-of-the-art in implementing quantum algorithms in practice). In particular, if there is a classical circuit for the classical oracle, there is a standard technique to obtain a quantum circuit of the same size which implement the corresponding quantum oracle and its inverse. Hence, our quantum algorithms have the potential to surpass blackbox classical algorithms in low dimensional settings where the oracle is given as an explicit circuit.

**Dimension dependence.**   Regarding potential concerns regarding the dependence of our quantum algorithms on the problem dimension, below we provide several supplementary points of context.

- As discussed, in the classical setting, prior research [27] demonstrated that when optimizing an 1-Lipschitz convex function, SGD has a query complexity of $\mathcal{O}(\epsilon^{-2})$ which is optimal, even in the one-dimensional case. In the quantum setting, we showed that, theoretically, quantum speedups which offer a different tradeoff between $\epsilon$ and $d$ dependencies are possible. Additionally, from our lower bound presented in Theorem 2 we know that some dimension dependence is inherent in obtaining an improvement.

- Classically, the complexity of dimension dependent optimization methods is well studied. In particular, there are parallel and private stochastic convex settings where the dimension dependencies are discussed, see e.g., [15, 18], and there exists works on critical point computation in low dimension settings, see e.g., [22].

- Even for high-dimensional problems, our algorithms can potentially be used as a subroutines for low-dimensional subproblems. For instance, in this paper we apply these method to the approximately best point problem (Problem 5), wherein we utilize our algorithm repeatedly within a one-dimensional setting.

### 1.3  Paper organization

In the next Section 2 we propose and develop a new algorithm for a new problem *quantum variance reduction*; this algorithm is the basis of our quantum speedups for stochastic optimization problems. We then discuss the application of quantum variance reduction for stochastic convex optimization by presenting two quantum algorithms in Section 3 and Section 4, respectively. Technically, it is possible to obtain our result in Section 4 just using the quantum mean estimation routine of [25] rather then quantum variance reduction, however our use of quantum variance reduction facilitates our presentation. Furthermore, we present quantum algorithms for non-convex optimization based on quantum variance reduction in Section 5. Finally, in Section 6 we prove quantum lower bounds for quantum variance reduction and quantum stochastic convex optimization that establish the optimality of our algorithms, and conclude the paper in Section 7.

## 2  Quantum variance reduction

We obtain our quantum speedups for stochastic optimization problems by proposing and developing new algorithms for a new problem which we call the *quantum variance reduction problem*.

**Problem 3** (Variance reduction). *For a $d$-dimensional random variable $X$ with $\mathrm{Var}[X] \leq L^2$ and some $\hat{\sigma} \geq 0$, suppose we are given access to its quantum sampling oracle $O_X$ defined in Definition 1. The goal is to output an unbiased estimate $\hat{\mu}$ of $\mu := \mathbb{E}[X]$ satisfying $\mathbb{E}\|\hat{\mu} - \mu\|^2 \leq \hat{\sigma}^2$.*

Classically, Problem 3 can be solved by averaging $\mathcal{O}(L^2/\hat{\sigma}^2)$ samples of $X$; this query complexity is optimal among classical algorithms [64]. However, if we have access to the *quantum sampling oracle* defined in Definition 1, [25] showed that a (possibly) biased estimate $\hat{\mu}$ with error $\|\hat{\mu} - \mu\| \leq \hat{\sigma}$ can be computed using $\widetilde{\mathcal{O}}(L\sqrt{d}\hat{\sigma}^{-1})$ queries.

**Lemma 1** ([25, Theorem 3.5]). *Given access to the quantum sampling oracle $O_X$, for any $\hat{\sigma}, \delta \geq 0$ there is a procedure* QuantumMeanEstimation $(X, \hat{\sigma}, \delta)$ *that uses $\widetilde{\mathcal{O}}(L\sqrt{d}\log(1/\delta)/\hat{\sigma})$ queries and outputs an estimate $\hat{\mu}$ of the expectation $\mu$ of any $d$-dimensional random variable $X$ satisfying $\mathrm{Var}[X] \leq L^2$ with error $\|\hat{\mu} - \mu\| \leq \hat{\sigma} \leq L$ and success probability $1 - \delta$.*

QuantumMeanEstimation [25] proceeds by introducing a directional mean function that reduces a multivariate mean estimation problem to a series of univariate mean estimation problem through quantum Fourier transform, which in the bounded norm case can be solved by quantum algorithms with a quadratic speedup using phase estimation. In terms of error rate, QuantumMeanEstimation in Lemma 1 improves over any classical sub-Gaussian estimator when $\frac{\hat{\sigma}}{L} \geq \frac{1}{\sqrt{d}}$. However, its bias hinders its combination with various optimization algorithms assuming unbiased inputs, see e.g., [57, 2, 29, 30]. In this work we show how to carefully combine their algorithm with a classic multi-level Monte-Carlo (MLMC) technique from [10, 7] to obtain an unbiased estimate $\hat{\mu}$ and success probability 1 with the same rate as [25] and prove the following theorem.

**Theorem 4.** *Algorithm 1 solves Problem 3 using an expected $\widetilde{\mathcal{O}}(Ld^{1/2}\hat{\sigma}^{-1})$ queries.*

In the following Table 2 we provide a comparison between our result and previous works and in Section 6 we prove that our algorithm is optimal up to a poly-logarithmic factor. Notably, our algorithm does not depend on the detailed implementation of QuantumMeanEstimation but only its query complexity. Hence, we believe that our approach may also be useful in removing the bias of other quantum mean estimation algorithms with similar expressions of query complexities, e.g., quantum phase estimation [55, 24] and quantum amplitude estimation [14].

Our algorithm consists of two components. First, we show that we can obtain a variant of the quantum mean estimation algorithm, denoted as QuantumMeanEstimation$^{+}$ $(X, \hat{\sigma})$, that outputs a low variance estimate with probability 1. This procedure compares the outcomes of the quantum estimator and a classical estimate, and in the event of significant disparity, generates a new independent classical estimate.

| Queries | Bias | Variance | Method |
|---------|------|----------|--------|
| $\hat{\sigma}^{-2}$ | $0$ | $\hat{\sigma}^2$ | Classical Variance Reduction |
| $d^{1/2}\hat{\sigma}^{-1}$ | $\hat{\sigma}$ | $\hat{\sigma}^2$ | Quantum Multivariate Mean estimation [25] |
| $\log^2\left(1/\gamma\right)\hat{\sigma}^{-1}$ | $\gamma$ | $\hat{\sigma}^2$ | One-dimensional Quantum Mean estimation [24] |
| $d^{1/2}\hat{\sigma}^{-1}$ | $0$ | $\hat{\sigma}^2$ | **Our Result** (Theorem 4) |

Table 2: Comparison between different methods for variance reduction in the case of $L = 1$ and $\hat{\sigma} \in (0, 1)$. The $\widetilde{\mathcal{O}}$ symbol was omitted in the "Queries" column.

Second, we use the MLMC technique [41], specifically a variant of the methods described in [10, 7], to carefully invoke the biased subroutine `QuantumMeanEstimation`$^+$ and compute the unbiased estimate. The algorithm, Algorithm 1, simply invokes `QuantumMeanEstimation`$^+$ for three randomly chosen accuracies and combines them to obtain the result. Though there are an infinite number of possible accuuracies chosen, we show that the expectation, the variance, and expected number of queries are all suitable to prove Theorem 4, whose proof is deferred to Appendix B.

---

**Algorithm 1:** Quantum variance reduction

---

**Input:** Random variable $X$, target variance $\hat{\sigma}^2$
**Output:** An unbiased estimate $\hat{\mu}$ of $\mathbb{E}[X]$ with variance at most $\hat{\sigma}^2$
1 Set $\tilde{\mu}_0 \leftarrow$ `QuantumMeanEstimation`$^+(X, \hat{\sigma}/10)$
2 Randomly sample $j \sim \text{Geom}\left(\frac{1}{2}\right) \in \mathbb{N}$
3 $\tilde{\mu}_j \leftarrow$ `QuantumMeanEstimation`$^+(X, 2^{-3j/4}\hat{\sigma}/10)$
4 $\tilde{\mu}_{j-1} \leftarrow$ `QuantumMeanEstimation`$^+(X, 2^{-3(j-1)/4}\hat{\sigma}/10)$
5 $\hat{\mu} \leftarrow \tilde{\mu}_0 + 2^j(\tilde{\mu}_j - \tilde{\mu}_{j-1})$
6 **return** $\hat{\mu}$

---

## 3 Quantum accelerated stochastic approximation

In this section, we present our $\widetilde{\mathcal{O}}(d^{5/8}\epsilon^{-3/2})$ query quantum algorithm for Problem 1. Our approach builds upon the framework proposed by Duchi et al. [28, 15], which involves performing a Gaussian convolution on the objective function $f$ and then optimizing the resulting smooth convoluted function. Compared to their algorithm, our algorithm differs by replacing the variance reduction step by our quantum variance reduction technique (Algorithm 1).

As with a variety of prior work on parallel and private SCO [28, 33, 15], we consider the smooth function $F_r$ that is the Gaussian convolution of the objective function $f$:

$$F_r(\mathbf{x}) := \int_{\mathbb{R}^d} \gamma_r(\mathbf{y})f(\mathbf{x} - \mathbf{y})\mathrm{d}\mathbf{y}, \quad \text{where} \quad \gamma_r(\mathbf{y}) := \frac{1}{(\sqrt{2\pi}r)^d}\exp\left(-\frac{\|\mathbf{y}\|^2}{2r^2}\right). \tag{5}$$

As shown in Lemma 4, when the radius $r$ of the convolution is sufficiently small, $F_r$ closely approximates $f$ pointwise. Consequently, to find an $\epsilon$-optimal point of $f$ it suffices to find an $\epsilon/4$-optimal point of $F_r$ for $r = \frac{\epsilon}{4\sqrt{d}L}$. Moreover, Lemma 4 shows that the stochastic gradient $\mathbf{g}_F$ of $F_r$ can be defined and obtained based on the stochastic gradient $\mathbf{g}$ of $f$ as follows:

$$\mathbf{g}_F(\mathbf{x}) = \mathbf{g}(\mathbf{x} - \mathbf{y}), \qquad \mathbf{y} \sim \gamma_r, \tag{6}$$

which satisfies

$$\underset{\mathbf{y}\sim\gamma_r}{\mathbb{E}}\,\mathbf{g}_F(\mathbf{x}) = \nabla F_r(\mathbf{x}) \quad \text{and} \quad \|\mathbf{g}_F(\mathbf{x})\| \leq L, \quad \forall\mathbf{x}.$$

Hence,

$$\|\nabla F_r(\mathbf{x})\| \leq \int_{\mathbf{y}\sim\gamma_r} \|\mathbf{g}_F(\mathbf{x})\|\mathrm{d}\mathbf{y} \leq L$$

indicating that $F_r$ is also $L$-Lipschitz.

To optimize this smooth convex function $F_r$, we leverage the accelerated stochastic approximation (AC-SA) algorithm introduced in [57], which applys an accelerated proximal descent method on the objective function using unbiased estimates of gradients. Our algorithm given in Algorithm 2 is a specialization of the AC-SA algorithm, where we implement those unbiased estimates of gradients using quantum variance reduction (Algorithm 1). In the classical setting, one query to the stochastic gradient $\mathbf{g}_F(\mathbf{x})$ of $F$ can be implemented by a random sampling a vector $\mathbf{y} \in \mathbb{R}^d$ from the Gaussian $\gamma_r$ followed by a query to the SGO $\mathcal{C}_\mathbf{g}$ (Definition 2) at $\mathbf{x} - \mathbf{y}$. Similarly, we can show that one query to a QSGO of $F_r$ (Definition 3) can also be implemented by one query to the QSGO of $f$. The subsequent theorem presents the query complexity of Algorithm 2.

---

**Algorithm 2:** Quantum accelerated stochastic approximation (Q-AC-SA)

**Input:** Function $f\colon \mathbb{R}^d \to \mathbb{R}$, precision $\epsilon$

**Parameters:** Domain Size $R$, total iteration budget $\mathcal{T} = \frac{4d^{1/4}LR}{\epsilon}$, target variance
$$\hat{\sigma} = \frac{d^{1/8}}{8}\sqrt{\frac{L\epsilon}{R}}, \text{ convolution radius } r = \frac{\epsilon}{4\sqrt{d}L}, \gamma = \frac{R\sqrt{6\ell_F}}{(\mathcal{T}+2)^{3/2}\hat{\sigma}}$$
**Output:** an $\epsilon$-optimal point of $F$

1 Denote $F_r(\mathbf{x}) := \int_{\mathbb{R}^d} \gamma_r(\mathbf{y})f(\mathbf{x} - \mathbf{y})\mathrm{d}\mathbf{y}$ as in (5)
2 Set $\mathbf{x}_1 \leftarrow \mathbf{0}, \mathbf{x}_1^{ag} \leftarrow \mathbf{x}_1$
3 **for** $t = 1, 2, \ldots, \mathcal{T}$ **do**
4 $\quad$ $\beta_t \leftarrow \frac{t+1}{2}, \gamma_t \leftarrow \frac{t+1}{2}\gamma$
5 $\quad$ $\mathbf{x}_t^{md} \leftarrow \beta_t^{-1}\mathbf{x}_t + (1 - \beta_t^{-1})\mathbf{x}_t^{ag}$
6 $\quad$ Call Algorithm 1 for an unbiased estimate $\tilde{\mathbf{g}}_t$ of $\nabla F_r(\mathbf{x}_t^{md})$ with variance at most $\hat{\sigma}^2$
7 $\quad$ $\mathbf{x}_{t+1} \leftarrow \underset{\mathbf{z} \in \mathbb{B}_R(\mathbf{0})}{\arg\min} \left\{ \gamma_t\langle\tilde{\mathbf{g}}_t, \mathbf{z} - \mathbf{x}_t^{md}\rangle + L\|\mathbf{x}_t^{md} - \mathbf{z}\|^2/(2r) \right\}$
8 $\quad$ $\mathbf{x}_{t+1}^{ag} = \beta_t^{-1}\mathbf{x}_{t+1} + (1 - \beta_t^{-1})\mathbf{x}_t^{ag}$
9 **return** $\mathbf{x}_{\mathcal{T}+1}^{ag}$

---

**Theorem 5** (Formal version of Theorem 1, Part 1). *Algorithm 2 solves Problem 1 using an expected* $\widetilde{\mathcal{O}}(d^{5/8}(LR/\epsilon)^{3/2})$ *queries.*

The proof of Theorem 5 is deferred to Appendix C.

## 4  Quantum stochastic cutting plane method (Q-SCP)

In this section, we develop our $\widetilde{\mathcal{O}}(d^{3/2}LR/\epsilon)$ query algorithm for Problem 1 which is based on a stochastic version of the cutting plane method. We introduce the key properties and related concepts of cutting plane methods, and then provide a procedure for efficiently post-processing the outcomes obtained from the stochastic cutting plane method using quantum variance reduction (Algorithm 1). Then, we analyze the overall query complexity for solving Problem 1. Technically, it is possible to obtain the results of this section using the quantum mean estimation routine of [25], rather then quantum variance reduction, however using quantum variance reduction facilitates our presentation.

We begin by introducing some notation and concepts on cutting plane methods. Cutting plane methods solve the *feasibility problem* defined as follows. Note that this problem is slightly easier to solve then the one in, e.g., [48], however it is simple suffices for our purposes.

**Problem 4** (Feasibility Problem). *We are given query access to a separation oracle for a set $K \subset \mathbb{R}^d$ such that on query $\mathbf{x} \in \mathbb{R}^d$ the oracle outputs a vector $\mathbf{c}$ and either $\mathbf{c} = \mathbf{0}$, in which case $\mathbf{x} \in K$, or $\mathbf{c} \neq \mathbf{0}$, in which case $H := \{\mathbf{z}\colon \mathbf{c}^\top\mathbf{z} \leq \mathbf{c}^\top\mathbf{x}\} \supset K$. The goal is to query a point $\mathbf{x} \in K$.*

[48] showed that Problem 4 can be solved by cutting plane method using $\mathcal{O}(d\log(dR/r))$ queries to a separation oracle where $R$ and $r$ denote bounds on $K$.

**Lemma 2** ([48, Theorem 1.1]). *There is a cutting plane method which solves Problem 4 using at most $C \cdot d\log(dR/r)$ queries for some constant $C$, given that the set $K$ is contained in the ball of radius $R$ centered at the origin and it contains a ball of radius $r$.*

[67, 58] demonstrated that, running cutting plane method on a convex function $f$ with the separation oracle being its gradient yields a sequence of points where at least one of them is an $\epsilon$-optimal point of $f$. This follows from the fact that there exists a ball of radius $\mathcal{O}(\epsilon)$ around $\mathbf{x}^*$ such that every point in this ball is $\epsilon$-optimal. In the stochastic setting, although we cannot access the precise gradient, we show that it suffices to use an $\mathcal{O}(\epsilon/R)$-approximate gradient oracle of $f$ (formally defined in Appendix D.1) as the separation oracle. Specifically, we prove the following result.

**Proposition 1.** *For any* $0 \leq \epsilon \leq LR$*, with success probability at least* $5/6$ *we can obtain* $\mathcal{T} = \mathcal{O}(d\log(dLR/\epsilon))$ *points* $\mathbf{x}_1, \ldots, \mathbf{x}_\mathcal{T} \in \mathbb{B}_R(\mathbf{0})$ *such that one of the* $x_i$ *is* $\epsilon$*-optimal using* $\widetilde{\mathcal{O}}\left(d^{3/2}LR/\epsilon\right)$ *queries to the QSGO* $O_\mathbf{g}$ *defined in Definition 3.*

The proof of Proposition 1 is deferred to Appendix D.1. After applying Proposition 1, it is not clear which query $\mathbf{x}_i$ is an $\mathcal{O}(\epsilon)$-optimal point. This difficulty arises because we lack access to the function value of $f$, which sets our problem apart from the feasibility problem discussed in [48], where there is a clear indication when a query successfully lies within the feasible region. Consequently, we next focus on identifying the optimal solution within the finite set $\Gamma$ of points using access to the QSGO. We conceptualize this task as the *approximately best point* problem, formulated as follows.

**Problem 5** (Approximately best point)**.** *For a L-Lipschitz convex function* $f: \mathbb{R}^d \rightarrow \mathbb{R}$ *and* $\mathcal{T}$ *points* $\mathbf{x}_1, \ldots, \mathbf{x}_\mathcal{T} \in \mathbb{B}_R(\mathbf{0})$*, find a convex combination* $\hat{\mathbf{x}} \in \mathbb{R}^d$ *of the points satisfying*

$$f(\hat{\mathbf{x}}) \leq \min_{t \in \mathcal{T}} f(\mathbf{x}_t) + \epsilon.$$

In this work, we develop an algorithm that solves Problem 5 by making pairwise comparisons in a hierarchical order, where each pairwise comparison is computed by running binary search on the segment along the segment between the two points. Formally, we prove the following result.

**Proposition 2.** *For any accuracy parameter* $\epsilon > 0$*, with success probability at least* $5/6$ *Algorithm 5 solves Problem 5 using* $\widetilde{\mathcal{O}}\left(RL\mathcal{T}/\epsilon\right)$ *queries to an L-bounded QSGO* $O_\mathbf{g}$ *defined in Definition 3.*

The proof of Proposition 2 and the corresponding quantum algorithm can be found in Appendix D.2. Next, we present the main result of this section, which describes the query complexity of solving Problem 1 using quantum stochastic cutting plane method.

**Corollary 1** (Formal version of Theorem 1, Part 2)**.** *With success probability at least* $2/3$*, Problem 1 can be solved using an expected* $\widetilde{\mathcal{O}}\left(d^{3/2}LR/\epsilon\right)$ *queries.*

The proof of Corollary 1 is deferred to Appendix D.3.

## 5 Quantum stochastic non-convex optimization

In this section, we present our quantum algorithms for Problem 2 in the bounded-variance setting and the mean-squared smoothness setting, respectively, using our quantum variance reduction technique.

To solve Problem 2 in the bounded variance setting, we leverage the randomized SGD method introduced in [40], which is a variant of SGD where the number of iterations is randomized. Our algorithm is a specialization of the randomized stochastic gradient algorithm, where we replace the classical variance reduction step by quantum variance reduction (Algorithm 1). The query complexity of our quantum algorithm is given in the following theorem.

**Theorem 6** (Formal version of Theorem 3, bounded variance setting)**.** *For any* $\epsilon > 0$*, Algorithm 6 solves Problem 2 in the bounded variance setting using an expected* $\widetilde{\mathcal{O}}(\Delta\ell\sigma\sqrt{d}\epsilon^{-3})$ *queries.*

The proof of Theorem 6 and the corresponding quantum algorithm can be found in Appendix E.

To solve Problem 2 in the mean-squared smoothness setting, we leverage the SPIDER algorithm introduced in [29], which is a variance reduction technique that allows us to estimate the gradient of a function with lower cost by utilizing the smoothness structure and reuse the stochastic gradient samples at nearby points. Our algorithm is a specialization of the SPIDER algorithm, where we replace the classical variance reduction step by quantum variance reduction (Algorithm 1). The query complexity of our quantum algorithm is given in the following theorem.

**Theorem 7** (Formal version of Theorem 3, mean-squared smoothness setting)**.** *For any* $0 \leq \epsilon \leq \sigma$*, Algorithm 7 solves Problem 2 in the mean-squared smoothness setting using an expected* $\widetilde{\mathcal{O}}\left(\ell\Delta\sqrt{d}\sigma\epsilon^{-2.5}\right)$ *number of queries.*

The proof of Theorem 7 and the corresponding quantum algorithm can be found in Appendix F.

## 6  Lower bounds

In this section we present two quantum lower bounds for solving quantum variance reduction (Problem 3) and stochastic convex optimization (Problem 1), respectively.

We first establish the following quantum lower bound for the variance reduction problem (Problem 3) which shows that our Algorithm 1 is optimal up to a poly-logarithmic factor when $\hat{\sigma} = \mathcal{O}\left(d^{-1/2}\right)$. By Markov's inequality, Proposition 3 equivalently states that any quantum algorithm that solves Problem 3 must make an expected $\Omega(L\sqrt{d}\hat{\sigma}^{-1})$ queries. This matches our algorithmic result provided in Theorem 4, up to a poly-logarithmic factor.

**Proposition 3.** *There is a constant $\alpha$ such that for any $\hat{\sigma} \leq \frac{\sigma}{\alpha\sqrt{d}}$, any quantum algorithm that solves Problem 3 with success probability at least $2/3$ must make at least $\Omega(L\sqrt{d}\hat{\sigma}^{-1})$ queries in the worst case.*

The proof of Proposition 3 is deferred to Appendix G.1

Next, we establish the following quantum lower bounds for stochastic convex optimization (Problem 1) in the low-dimension regime and the high-dimension regime, respectively, which show that our quantum stochastic cutting plane method in Section 4 is optimal up to a poly-logarithmic factor when the dimension $d$ is a constant, and there is no quantum speedup over SGD when $d \geq \Omega\left(\epsilon^{-2}\right)$.

**Theorem 8.** *For any $\epsilon \leq \frac{RL}{100\sqrt{d}}$, any quantum algorithm that solves Problem 1 with success probability at least $2/3$ must make at least $\Omega(\sqrt{d}RL/\epsilon)$ queries in the worst case. For any $\frac{RL}{100\sqrt{d}} \leq \epsilon \leq 1$, any quantum algorithm that solves Problem 1 with success probability at least $2/3$ must make at least $\Omega(R^2L^2/\epsilon^2)$ queries in the worst case.*

The proof of Theorem 8 is deferred to Appendix G.2.

## 7  Conclusion

We presented improved quantum algorithms for stochastic optimization. We developed a new technical tool which we call *quantum variance reduction* and show how to use it to improve upon the query complexity for stochastic convex optimization and for critical point computation in smooth, stochastic, non-convex functions. Further, we provided lower bounds which establish both the optimality of our quantum variance reduction technique and of one of our stochastic convex optimization algorithms in low dimensions. A natural open problem suggested by our work is to establish the optimal complexity of the problems we study, e.g., stochastic convex optimization and stochastic non-convex optimization with quantum oracle access, in higher dimensions. We hope this paper fuels further study of these problems.

## Acknowledgement

We thank Adam Bouland, Yair Carmon, András Gilyén, Yujia Jin, and Tongyang Li for helpful discussions. A.S. was supported in part by a Microsoft Research Faculty Fellowship, NSF CAREER Award CCF-1844855, NSF Grant CCF-1955039, a PayPal research award, and a Sloan Research Fellowship. C.Z. was supported in part by the Shoucheng Zhang Graduate Fellowship.

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

# A  Additional related work

**Stochastic convex optimization.**   Stochastic convex optimization is a broad, well-studied area of research. Beyond the works mentioned in the introduction, for additional references and discussion of research in the area, see [11, 47, 68] for detailed overviews of stochastic convex optimization methods and [19] for a presentation of convex optimization in the online learning setting. Additionally, see [26, 51] for a discussion of the closely related problem of finite-sum optimization.

**Non-convex optimization.**   Non-convex optimization is a rapidly advancing research area in optimization theory. These advances are motivated in part by the fact that the landscapes of various modern machine learning problems are typically non-convex, including deep neural networks, principal component analysis, tensor decomposition, etc. In general, finding a global optimum of a non-convex function is NP-hard [66, 69]. Hence, many theoretical works instead focus on finding local minima rather than a global one, given existing empirical and theoretical evidence that local minima can be as good as global minima in certain machine learning problems [8, 34, 36, 37, 39, 46]. The first step of finding local minima would be to find stationary points, which has been discussed in many previous works [1, 9, 17, 16, 70, 71]. An important line of work focuses on first-order, (stochastic) gradient-based algorithms for finding critical points and local minima since higher-order derivatives are often not accessible in practical scenarios [3, 17, 29, 30, 34, 49, 50, 62, 78, 81]. There are also results on non-convex optimization that study different settings [38, 61, 79, 85, 35, 83, 84].

**Quantum Monte Carlo methods.**   Since [65], quantum Monte Carlo methods have been broadly investigated. In particular, [59] discussed quantum Monte-Carlo methods for entropy estimation, and [45] initiated the study of quantum mean estimation problem with multiplicative error, which is followed by [44, 25, 24, 56]. Multilevel Monte Carlo methods are also widely used in quantum algorithms, see e.g., [4, 60]. Specifically, the particular multilevel Monte Carlo strategies we use for quantum variance reduction have roots in the classic computation work of [10, 7].

# B  Proof of Theorem 4

---

**Algorithm 3:** `QuantumMeanEstimation`$^+(X, \hat{\sigma})$

---

**Input:** Random variable $X$, target variance $\hat{\sigma}^2 \le L^2$
**Parameters:** $\delta = \hat{\sigma}^6/(4L)^6$, $D = \frac{\hat{\sigma}}{4} + \frac{16L^3}{\hat{\sigma}^2}$
**Output:** An estimate $\hat{\mu}$ of $\mu = \mathbb{E}[X]$ satisfying $\mathbb{E}\|\hat{\mu} - \mu\|^2 \le \hat{\sigma}^2$
1  Set $X_1 \leftarrow$ `QuantumMeanEstimation`$(X, \hat{\sigma}/4, \delta)$
2  Randomly draw one classical sample $X_2$ of $X$
3  **if** $\|X_1 - X_2\| \le D$ **then return** $X_1$
4  **else**
5  | Randomly draw one classical sample $X_3$ of $X$
6  | **return** $X_3$

---

We begin by presenting a lemma showing how two unbiased, bounded variance random variables ($X_2$ and $X_3$ in the lemma) can be used to control the expected $\ell_2$ error of a random variable ($X_1$ in the lemma) that is close to the expectation of $X_2$ and $X_3$ with some probability. In Algorithm 3 $X_1$ corresponds to the output of `QuantumMeanEstimation` and then $X_2$ and $X_3$ are obtained by classic sampling.

**Lemma 3.** *Let $X_1, X_2, X_3 \in \mathbb{R}^d$ be independent random variables where $\mathbb{E}X_2 = \mathbb{E}X_3 = \mu$, $\text{Var}[X_2] \le L^2$, and $\text{Var}[X_3] \le L^2$, and $\|X_1 - \mu\| \le \hat{\sigma}$ with probability at least $1 - \delta$. Further, let $Z$ be the random variable that is $X_1$ if $\|X_1 - X_2\| \le D$ and $X_3$ otherwise where $D > \hat{\sigma}$. Then,*

$$\mathbb{E}\|Z - \mu\|^2 \le \hat{\sigma}^2 + L\left(\frac{L^2 - \hat{\sigma}^2}{D - \hat{\sigma}}\right) + \delta(2D^2 + 3L^2).$$

*Consequently, $\mathbb{E}\|Z - \mu\|^2 \le 13\sigma^2$ when $D = \hat{\sigma} + L^3/\hat{\sigma}^2$ and $\delta \le \hat{\sigma}^6/L^6$.*

*Proof.* Let $S$ denote the event that $\|X_1 - \mu\|_2 \leq \hat{\sigma}$ and let $T$ denote the event that $\|X_1 - X_2\|_2 \leq D$. Then by the law of total expectation

$$\mathbb{E}\|Z - \mu\|^2 = \Pr[S]\,\mathbb{E}\left[\|Z - \mu\|_2^2 \mid S\right] + \Pr[\text{not } S]\,\mathbb{E}\left[\|Z - \mu\|_2^2 \mid \text{not } S\right]$$

$$\leq \mathbb{E}\left[\|Z - \mu\|^2 \mid S\right] + \delta \cdot \mathbb{E}\left[\|Z - \mu\|^2 \mid \text{not } S\right]$$

Where we used that $\Pr[S] \leq 1$ and $\Pr[\text{not } S] \leq \delta$. We will prove the lemma by leveraging this inequality and upper bounding both $\mathbb{E}\left[\|Z - \mu\|_2^2 \mid S\right]$ and $\mathbb{E}\left[\|Z - \mu\|_2^2 \mid \text{not } S\right]$.

First, we upper bound $\mathbb{E}\left[\|Z - \mu\|_2^2 \mid S\right]$. Again by the law of total expectation and the definition of $Z$ we have that

$$\mathbb{E}\left[\|Z - \mu\|_2^2 \mid S\right] = \Pr[T \mid S]\,\mathbb{E}\left[\|Z - \mu\|_2^2 \mid T \text{ and } S\right] \tag{7}$$

$$+ \Pr[\text{not } T \mid S]\,\mathbb{E}\left[\|Z - \mu\|_2^2 \mid T \text{ and not } S\right]$$

$$\leq \Pr[T \mid S] \cdot \hat{\sigma}^2 + \Pr[\text{not } T \mid S]\,L^2. \tag{8}$$

Here we use that if $T$ and $S$ both hold then $Z = X_1$ and $\|X_1 - \mu\| \leq \hat{\sigma}$ and that if $S$ holds and not $T$ then $Z = X_3$ and $X_3$ is independent of $S$ and $T$ with variance $L^2$. Now, if $\|X_2 - \mu\|_2 \leq D - \hat{\sigma}$ then when $S$ holds by triangle inequality

$$\|X_1 - X_2\| \leq \|X_1 - \mu\| + \|X_2 - \mu\| \leq D.$$

Applying this bound and Chebyshev inequality to $X_2$ (and using that $X_2$ is independent of $S$) we have that

$$\Pr[T \mid S] \geq \Pr[\|X_2 - \mu\|_2 \leq D - \hat{\sigma} \mid S] = \Pr[\|X_2 - \mu\|_2 \leq D - \hat{\sigma}] \geq 1 - \frac{L}{D - \hat{\sigma}}.$$

Since $\hat{\sigma} < L$, combined with (7) this implies that

$$\mathbb{E}\left[\|Z - \mu\|_2^2 \mid S\right] \leq \left(1 - \frac{L}{D - \hat{\sigma}}\right)\hat{\sigma}^2 + \frac{L^3}{D - \hat{\sigma}} = \hat{\sigma}^2 + L\left(\frac{L^2 - \hat{\sigma}^2}{D - \hat{\sigma}}\right)$$

Next, we upper bound $\mathbb{E}\left[\|Z - \mu\|_2^2 \mid \text{not } S\right]$. Note that if $T$ holds then

$$\|Z - \mu\|^2 \leq 2\|Z - X_2\|^2 + 2\|X_2 - \mu\|^2 \leq 2D^2 + 2\|X_2 - \mu\|^2.$$

Since $Z = X_3$ if $T$ does note hold, we then see that in either case

$$\|Z - \mu\|^2 \leq 2D^2 + 2\|X_2 - \mu\|^2 + \|X_3 - \mu\|^2$$

Since, $X_2$ and $X_3$ are independent of $S$ with variance $L$ this therefore implies that

$$\mathbb{E}\left[\|Z - \mu\|_2^2 \mid \text{not } S\right] \leq \mathbb{E}\left[2D^2 + 2\|X_2 - \mu\|^2 + \|X_3 - \mu\|^2\right] = 2D^2 + 3L^2,$$

which leads to

$$\mathbb{E}\|Z - \mu\|^2 \leq \mathbb{E}\left[\|Z - \mu\|^2 \mid S\right] + \delta \cdot \mathbb{E}\left[\|Z - \mu\|^2 \mid \text{not } S\right]$$

$$\leq \hat{\sigma}^2 + L\left(\frac{L^2 - \hat{\sigma}^2}{D - \hat{\sigma}}\right) + \delta(2D^2 + 3L^2).$$

$\square$

**Corollary 2.** *Given access to the quantum sampling oracle $O_X$ of a $d$-dimensional random variable $X$, for any $\hat{\sigma}, \delta \geq 0$, the procedure* `QuantumMeanEstimation`$^+$ $(X, \hat{\sigma})$ *(Algorithm 3) uses $\widetilde{\mathcal{O}}(L\sqrt{d}\hat{\sigma}^{-1})$ queries and outputs an estimate $\hat{\mu}$ of the expectation $\mu = \mathbb{E}[X]$ satisfying $\mathbb{E}\|\hat{\mu} - \mu\|^2 \leq \hat{\sigma}^2$.*[4]

---

[4]At Line 2 and Line 5 of Algorithm 3, we can further refine our approach to obtain $X_2$ and $X_4$ by taking the same number of classical samples as we would have used in the `QuantumMeanEstimation` procedure, which can reduce the poly-logarithmic factor in the overall query complexity.

*Proof.* By Lemma 3, the output $\hat{\mu}$ of QuantumMeanEstimation$^+$ $(X, \hat{\sigma})$ (Algorithm 3) satisfies

$$\mathbb{E}\|\hat{\mu} - \mu\|^2 \leq \hat{\sigma}^2. \tag{9}$$

As for the number of queries to $O_X$, note that it uses

$$\widetilde{\mathcal{O}}\left(\frac{L\sqrt{d}\log(1/\delta)}{\hat{\sigma}}\right) = \widetilde{\mathcal{O}}\left(\frac{L\sqrt{d}}{\hat{\sigma}}\log\left(\frac{L^6}{\hat{\sigma}^6}\right)\right) = \widetilde{\mathcal{O}}\left(\frac{L\sqrt{d}}{\hat{\sigma}}\right)$$

queries to prepare $X_1$. Moreover, we can prepare the classical samples $X_2$ and $X_3$ by measuring the quantum state $O_X\,|0\rangle$ in the computational basis, which takes 1 query to $O_X$. Hence, the overall query complexity equals $\widetilde{\mathcal{O}}(L\sqrt{d}\hat{\sigma}^{-1})$. $\qquad\square$

Equipped with Corollary 2, we are ready to prove Theorem 4.

*Proof of Theorem 4.* The structure of our proof is similar to the proof of [7, Proposition 1]. Observe that the output $\hat{\mu}$ of Algorithm 1 can be expressed as

$$\hat{\mu} = \tilde{\mu}_0 + 2^J(\tilde{\mu}_J - \tilde{\mu}_{J-1}), \qquad J \sim \mathrm{Geom}\left(\frac{1}{2}\right) \in \mathbb{N}. \tag{10}$$

Then we have

$$\mathbb{E}[\hat{\mu}] = \mathbb{E}[\tilde{\mu}_0] + \sum_{j=1}^{\infty} \Pr\{J = j\}2^j\left(\mathbb{E}[\tilde{\mu}_j] - \mathbb{E}[\tilde{\mu}_{j-1}]\right) = \mathbb{E}[\tilde{\mu}_\infty] = \mu,$$

given that $\Pr\{J = j\} = 2^{-j}$. As for the variance, we have

$$\mathbb{E}\|\hat{\mu} - \mu\|^2 \leq 2\mathbb{E}\|\hat{\mu} - \tilde{\mu}_0\|^2 + 2\mathbb{E}\|\tilde{\mu}_0 - \mu\|^2,$$

where

$$\mathbb{E}\|\hat{\mu} - \tilde{\mu}_0\|^2 = \sum_{j=1}^{\infty} \Pr(J = j)2^{2j}\mathbb{E}\|\tilde{\mu}_j - \tilde{\mu}_{j-1}\|^2 = \sum_{j=1}^{\infty} 2^j\mathbb{E}\|\tilde{\mu}_j - \tilde{\mu}_{j-1}\|^2,$$

and for each $j$ we have

$$\mathbb{E}\|\tilde{\mu}_j - \tilde{\mu}_{j-1}\|^2 \leq 2\mathbb{E}\|\tilde{\mu}_j - \mu\|^2 + 2\mathbb{E}\|\tilde{\mu}_{j-1} - \mu\|^2.$$

By Corollary 2,

$$\mathbb{E}\|\tilde{\mu}_j - \mu\|^2 \leq \frac{\hat{\sigma}^2}{100 \cdot 2^{3j/2}}, \quad \forall j \geq 0,$$

which leads to

$$\mathbb{E}\|\tilde{\mu}_j - \tilde{\mu}_{j-1}\|^2 \leq \frac{\sigma^2}{50 \cdot 2^{3(j-1)/2}} + \frac{\sigma^2}{50 \cdot 2^{3j/2}} \leq \frac{\hat{\sigma}^2}{10 \cdot 2^{3j/2}},$$

and

$$\mathbb{E}\|\hat{\mu} - \tilde{\mu}_0\|^2 = \frac{\hat{\sigma}^2}{10}\sum_{j=1}^{\infty} \frac{1}{2^{j/2}} \leq \frac{1}{3}\hat{\sigma}^2.$$

Hence,

$$\mathbb{E}\|\hat{\mu} - \mu\|^2 \leq 2\mathbb{E}\|\hat{\mu} - \tilde{\mu}_0\|^2 + 2\mathbb{E}\|\tilde{\mu}_0 - \mu\|^2 \leq \hat{\sigma}^2,$$

given that $\mathbb{E}\|\tilde{\mu}_0 - \mu\|^2 \leq \hat{\sigma}^2/100$ by Corollary 2. Moreover, the expected number of queries is

$$\widetilde{\mathcal{O}}\left(\frac{L\sqrt{d}\log d}{\hat{\sigma}}\right) \cdot \left(1 + \sum_{j=1}^{\infty} \Pr\{J = j\} \cdot \left(2^{3j/4} + 2^{3(j-1)/4}\right)\right) = \widetilde{\mathcal{O}}\left(\frac{L\sqrt{d}}{\hat{\sigma}}\right).$$

$\square$

## C   Proof of Theorem 5

We first state a result from [15] which bounds properties of $F$ defined in (5). The lemma states that, provided the radius $r$ of the convolution is sufficiently small, $F$ is pointwise close to $f$ and hence finding an $\mathcal{O}(\epsilon)$-minimum of $f$ is equivalent to finding an $\epsilon$-minimum of $F$. Further the lemma shows that $F$ is Lipschitz and smooth.

**Lemma 4** ([15, Lemma 8]). *For any $L$-Lipschitz convex function $f \colon \mathbb{R}^d \to \mathbb{R}$, its Gaussian convolution $g$ defined in Eq. (5) is convex, $L$-Lipschitz, and satisfies*

$$|f(\mathbf{x}) - g(\mathbf{x})| \le \sqrt{d} \cdot Lr \quad and \quad \nabla^2 g(\mathbf{x}) \le (L/r) \cdot \mathbb{I}_d, \quad \forall \mathbf{x} \in \mathbb{R}^d,$$

*where $\mathbb{I}_d$ is the $d$-dimensional identity matrix.*

Prior to proving Theorem 5, we show that one query to a QSGO $O_{\mathbf{g}_F}$ (Definition 3) of $F_r$ satisfying

$$O_{\mathbf{g}_F} |\mathbf{x}\rangle \otimes |0\rangle \otimes |0\rangle \to |\mathbf{x}\rangle \otimes \int_{\mathbf{v} \in \mathbb{R}^d} \sqrt{p_{F,\mathbf{x}}(\mathbf{v}) \mathrm{d}\mathbf{v}} \, |\mathbf{v}\rangle \otimes |\mathrm{garbage}(\mathbf{v})\rangle, \tag{11}$$

can be implemented by one query to $O_{\mathbf{g}}$ of $f$ defined in Definition 3.

**Lemma 5.** *The QSGO $O_{\mathbf{g}_F}$ of $F_r$ defined in (11) can be implemented with one query to the QSGO $O_{\mathbf{g}}$ of $f$.*

*Proof.* We first prepare the following quantum state

$$|\psi\rangle = |\mathbf{x}\rangle \otimes \left( \int_{\mathbf{y} \in \mathbb{R}^d} \sqrt{\gamma_r(\mathbf{y}) \mathrm{d}\mathbf{y}} \, |\mathbf{x} - \mathbf{y}\rangle \right) \otimes |0\rangle, \quad \forall \mathbf{x} \in \mathbb{R}^d.$$

Applying $O_{\mathbf{g}}$ to the last two registers yields

$$(\mathbb{I} \otimes O_{\mathbf{g}}) |\psi\rangle = |\mathbf{x}\rangle \otimes \left( \int_{\mathbf{y} \in \mathbb{R}^d} \sqrt{\gamma_r(\mathbf{y}) \mathrm{d}\mathbf{y}} \, |\mathbf{x} - \mathbf{y}\rangle \otimes \int_{\mathbf{v} \in \mathbb{R}^d} \sqrt{p_{f,\mathbf{x}-\mathbf{y}}(\mathbf{v}) \mathrm{d}\mathbf{v}} \, |\mathbf{v}\rangle \right)$$

$$= |\mathbf{x}\rangle \otimes \int_{\mathbf{y},\mathbf{v} \in \mathbb{R}^d} \sqrt{\gamma_r(\mathbf{y}) p_{f,\mathbf{x}-\mathbf{y}}(\mathbf{v}) \mathrm{d}\mathbf{y} \mathrm{d}\mathbf{v}} \, |\mathbf{x} - \mathbf{y}\rangle |\mathbf{v}\rangle. \tag{12}$$

Given that

$$p_{F,\mathbf{x}}(\mathbf{v}) = \int_{\mathbf{y} \in \mathbb{R}^d} \gamma_r(\mathbf{y}) p_{f,\mathbf{x}-\mathbf{y}}(\mathbf{v}) \mathrm{d}\mathbf{y}, \qquad \forall \mathbf{v} \in \mathbb{R}^d,$$

if we measure the last register, the probability density function of the outcome would be exactly $p_{F,\mathbf{x}}$. Hence, the quantum state $(\mathbb{I} \otimes O_{\mathbf{g}}) |\psi\rangle$ in (12) can also be written as

$$(\mathbb{I} \otimes O_{\mathbf{g}}) |\psi\rangle = |\mathbf{x}\rangle \otimes \int_{\mathbf{v} \in \mathbb{R}^d} \sqrt{p_{F,\mathbf{x}}(\mathbf{v}) \mathrm{d}\mathbf{v}} \, |\mathrm{garbage}(\mathbf{v})\rangle \otimes |\mathbf{v}\rangle.$$

By swapping the last two quantum registers, we can obtain the desired output state of $O_{\mathbf{g}_F}$. $\qquad \square$

The following result from [57] bounds the rate at which Algorithm 2 decreases the function error of $F_r$. Note that validity of this result relies solely on the fact that, at Line 6, the variance of the unbiased gradient estimate $\tilde{\mathbf{g}}_t$ does not exceed $\hat{\sigma}^2$, irrespective of its implementation.

**Lemma 6.** *The output $\mathbf{x}_{\mathcal{T}+1}^{ag}$ of Algorithm 2 satisfies*

$$\mathbb{E}[F_r(\mathbf{x}_{\mathcal{T}+1}^{ag}) - F_r^*] \le \frac{4LR^2}{r\mathcal{T}(\mathcal{T}+2)} + \frac{4R\hat{\sigma}}{\sqrt{\mathcal{T}}},$$

*where $F_r$ is the convoluted function defined in Line 1 of Algorithm 2 and $F_r^*$ is its minimum.*

*Proof.* This lemma follows from [57, Corollary 1], which shows that

$$\mathbb{E}[F_r(\mathbf{x}_{\mathcal{T}+1}^{ag}) - F_r^*] \le \frac{4\ell_F R^2}{\mathcal{T}(\mathcal{T}+2)} + \frac{4R\hat{\sigma}}{\sqrt{\mathcal{T}}},$$

where $\ell_F$ is the Lipschitz parameter of $F_r$, which equals $L/r$ by Lemma 4. $\qquad \square$

Equipped with above results, we are now ready to present the proof of Theorem 5.

*Proof of Theorem 5.* By Lemma 6, the output $\mathbf{x}_{\text{out}}$ of Algorithm 2 satisfies

$$\mathbb{E}[F(\mathbf{x}_{\text{out}}) - F^*] \leq \frac{4\ell_F R^2}{r\mathcal{T}(\mathcal{T}+2)} + \frac{4R\hat{\sigma}}{\sqrt{\mathcal{T}}} \leq \frac{\epsilon}{2}. \tag{13}$$

Moreover, by Lemma 4 we can derive that

$$\mathbb{E}[f(\mathbf{x}_{\text{out}}) - f^*] \leq \mathbb{E}[f(\mathbf{x}_{\text{out}}) - F(\mathbf{x}_{\text{out}})] + (f^* - F^*) + \mathbb{E}[F(\mathbf{x}_{\text{out}}) - F^*] = \epsilon.$$

Finally we bound the expected number of queries to $O_{\mathbf{g}}$ used in the algorithm. Since $\|\mathbf{g}_F(\mathbf{x})\| \leq L$ for any $\mathbf{x} \in \mathbb{R}^d$, by Theorem 4 we know that Algorithm 1 can output an estimate $\tilde{\mathbf{g}}_F(\mathbf{x}_t^{md})$ of $\nabla F(\mathbf{x}_t^{md})$ with variance at most $\hat{\sigma}^2$ using $\widetilde{\mathcal{O}}(L\sqrt{d}/\hat{\sigma})$ queries to the oracle $O_F$, and hence the same number of queries to $O_{\mathbf{g}}$ by Lemma 5. Then, the total number of queries to $O_{\mathbf{g}}$ equals

$$\mathcal{T} \cdot \widetilde{\mathcal{O}}\left(\frac{L\sqrt{d}}{\hat{\sigma}}\right) = \frac{d^{1/4}LR}{\epsilon} \cdot \widetilde{\mathcal{O}}\left(\frac{L\sqrt{d}}{d^{1/8}} \cdot \sqrt{\frac{R}{L\epsilon}}\right) = \widetilde{\mathcal{O}}\left(d^{5/8} \cdot \left(\frac{LR}{\epsilon}\right)^{3/2}\right).$$

$\square$

# D Proof of quantum stochastic cutting plane method

## D.1 Proof of Proposition 1

**Definition 5** ($\delta$-approximate gradient oracle ($\delta$-AGO)). *For $f \colon \mathbb{R}^d \to \mathbb{R}$, its $\delta$-approximate gradient oracle, $\delta$-AGO, is defined as a random function that when queried at $\mathbf{x}$, returns a vector $\tilde{\mathbf{g}}(\mathbf{x})$ that satisfies $\|\tilde{\mathbf{g}}(\mathbf{x}) - \nabla f(\mathbf{x})\| \leq \delta$.*

We show that a $\delta$-AGO can be efficiently implemented using our quantum variance reduction algorithm. To obtain this result we can also use [53, Claim 1] by changing some parameters. Nevertheless, compared to their deterministic implementation, our approach has a lower expected time complexity (though is randomized).

**Lemma 7.** *For any $\delta, \xi \geq 0$, with success probability at least $1-\xi$, the $\delta$-AGO of a Lipschitz function $f$ can be implemented using $\mathcal{O}(\log(1/\xi))$ calls to Algorithm 1 with in total $\mathcal{O}(L\sqrt{d}\log(1/\xi)/\delta)$ queries to the QSGO.*

*Proof.* We implement the $\delta$-AGO using the following procedure. First, we obtain $k = \log(1/\xi)+10$ independent unbiased estimates $\mathbf{g}_1, \ldots, \mathbf{g}_k$ of $\nabla f(\mathbf{x})$ with variance $\delta^2/16$ by Algorithm 1, using

$$k \cdot \frac{L\sqrt{d}}{\delta} = \mathcal{O}(L\sqrt{d}\log(1/\xi)/\delta)$$

queries in total to the QSGO $O_{\mathbf{g}}$, by Theorem 4. Note that any individual sample $\mathbf{g}_i$ satisfies

$$\Pr\left[\|\mathbf{g}_i - \nabla f(\mathbf{x})\| \geq \frac{\delta}{2}\right] \leq \frac{1}{4}.$$

Hence, if we let $S$ denote the subset of $[k]$ where each $i \in S$ have $\ell_2$ distance to $\nabla f(\mathbf{x})$ at most $\delta/2$, then by Hoeffding's inequality, $|S| \geq 2k/3$ with probability at least

$$1 - \exp\left(-\frac{2(k/3 - k/4)^2}{k}\right) \geq 1 - \xi.$$

Observe that for any sample $\mathbf{g}_i$ satisfying $\|\mathbf{g}_i - \mathbf{g}_j\| \leq \delta/2$ for at least $2k/3$ of $\mathbf{g}_j$'s, there must exist at least some $\mathbf{g}_{j'} \in S$ such that $\|\mathbf{g}_i - \mathbf{g}_{j'}\| \leq \delta/2$, and therefore $\|\mathbf{g}_i - \nabla f(\mathbf{x})\| \leq \delta$ by triangle inequality. Consequently, we just need to find and output any such $\mathbf{g}_i$ as an $\delta$-AGO, which is guaranteed to exist given that any sample in $S$ satisfies this property. This can be done in expected $\mathcal{O}(k)$ time by picking a random estimate and checking whether it is close to at least $2k/3$ of other estimates. $\square$

Next, we establish that we can query an $\epsilon$-optimal point by applying the cutting plane method with the separation oracle being an $\mathcal{O}(\epsilon/R)$-approximate gradient oracle, as provided by Lemma 7.

*Proof of Proposition 1.* Define $K_{\epsilon/2}$ as the set of $\epsilon/2$-optimal points of the function $f$, and $K_\epsilon$ as the set of $\epsilon$-optimal points of $f$. We know that $K_{\epsilon/2}$ contains a ball of radius at least $r_K = \epsilon/(2L)$ since for any $\mathbf{x}$ with $\|\mathbf{x} - \mathbf{x}^*\| \le r_K$ we have

$$f(\mathbf{x}) - f(\mathbf{x}^*) \le L\|\mathbf{x} - \mathbf{x}^*\| \le \frac{\epsilon}{2}. \tag{14}$$

We apply the cutting plane method, as described in Lemma 7, to query a point in $K_{\epsilon/2}$, which is a subset of the ball $\mathbb{B}_{2R}(\mathbf{0})$. To achieve this, we use an $\epsilon/(10R)$-approximate gradient oracle ($\epsilon/(10R)$-AGO) of $f$, implemented using Lemma 7, as the separation oracle for the cutting plane method. Throughout this process, we assume each query to the $\epsilon/(10R)$-AGO is successfully executed, and we will later discuss the error probability associated with this assumption. In this case, we show that any query outside of $K_\epsilon$ to the $\epsilon/(10R)$-AGO will be a valid separation oracle for $K_{\epsilon/2}$. In particular, if we ever queried the $\epsilon/(10R)$-AGO at any $\mathbf{x} \in \mathbb{B}_{2R}(\mathbf{0}) \setminus K_\epsilon$ with output being $\tilde{\mathbf{g}}$, for any $\mathbf{y} \in K_{\epsilon/2}$ we have

$$\begin{aligned}
\langle \tilde{\mathbf{g}}, \mathbf{y} - \mathbf{x} \rangle &\le \langle \nabla f(\mathbf{x}), \mathbf{y} - \mathbf{x} \rangle + \|\tilde{\mathbf{g}} - \nabla f(\mathbf{x})\| \cdot \|\mathbf{y} - \mathbf{x}\| \\
&\le f(\mathbf{y}) - f(\mathbf{x}) + \|\tilde{\mathbf{g}} - \nabla f(\mathbf{x})\| \cdot \|\mathbf{y} - \mathbf{x}\| \\
&\le -\frac{\epsilon}{2} + \frac{\epsilon}{10R} \cdot 4R < 0,
\end{aligned}$$

where the second inequality is due to the convexity of $f$, indicating that $\tilde{\mathbf{g}}$ is a valid separation oracle for the set $K_{\epsilon/2}$. Consequently, upon applying Lemma 2, we can deduce that after $Cd\log(dR/r_K)$ iterations, at least one of the queries must lie within $K_\epsilon$.

To ensure an overall success probability of at least $5/6$, we employ the union bound, which necessitates that each query to the $\epsilon/(10R)$-AGO be implemented with a failure probability of no more than $(6Cd\log(R/r_K))^{-1}$, which by Lemma 7 requires

$$\mathcal{O}\left( \frac{10LR\sqrt{d}}{\epsilon} \log(6Cd\log(dR/r_K)) \right) = \widetilde{\mathcal{O}}\left( \frac{LR\sqrt{d}}{\epsilon} \right)$$

queries to the QSGO $O_{\mathbf{g}}$, and the overall query complexity equals

$$\widetilde{\mathcal{O}}\left( \frac{LR\sqrt{d}}{\epsilon} \right) \cdot Cd\log(dR/r_K) = \widetilde{\mathcal{O}}\left( \frac{d^{3/2}LR}{\epsilon} \right).$$

$\square$

### D.2  Proof of Proposition 2

We first present the following algorithm that makes pairwise comparison using queries to the QSGO(Definition 3).

Note that in Line 7 of Algorithm 4 the estimation step is carried out in a projected space along the vector $\hat{\mathbf{e}}$. This is motivated by the fact that we are specifically interested in the information of the gradient $\nabla f$ within this projected space, which effectively reduces to a one-dimensional variable. As a result, the mean estimation can be performed without introducing an additional factor of $\sqrt{d}$ in the query complexity as given in Theorem 4.

**Lemma 8.** *For any $\mathbf{y}_{l0}, \mathbf{y}_{r0} \in \mathbb{B}_R(\mathbf{0})$ and any $\epsilon' > 0$, with success probability at least $1/(6\mathcal{T})$ Algorithm 4 returns a point $\hat{\mathbf{y}} \in \mathbb{R}^d$ satisfying*

$$f(\hat{\mathbf{y}}) \le \min_{\lambda \in [0,1]} f\left( \lambda\mathbf{y}_{l0} + (1-\lambda)\mathbf{y}_{r0} \right) + \epsilon'$$

*using $\widetilde{\mathcal{O}}\left( RL/\epsilon' \right)$ queries to an $L$-bounded QSGO $O_{\mathbf{g}}$ defined in Definition 3.*

---

**Algorithm 4:** `StochasticLineSearch(`$\mathbf{y}_{l0}, \mathbf{y}_{r0}, \epsilon'$`)`

---

**Input:** Endpoints $\mathbf{y}_l, \mathbf{y}_r \in \mathbb{B}_R(\mathbf{0})$, accuracy $\epsilon'$
**Output:** $\hat{\mathbf{y}}$ such that $f(\hat{\mathbf{y}}) \leq \min_{\lambda \in [0,1]} f(\lambda \mathbf{y}_l + (1-\lambda)\mathbf{y}_r) + \epsilon'$.

**1** $\mathbf{y}_l \leftarrow \mathbf{y}_{l0}, \mathbf{y}_r \leftarrow \mathbf{y}_{r0}$
**2** **if** $\mathbf{y}_l = \mathbf{y}_r$ **then return** $\mathbf{y}_l$
**3** **else**
**4**     $\hat{\mathbf{e}} \leftarrow \frac{\mathbf{y}_r - \mathbf{y}_l}{\|\mathbf{y}_r - \mathbf{y}_l\|}$
**5**     **repeat**
**6**         $\mathbf{y}_m \leftarrow (\mathbf{y}_l + \mathbf{y}_r)/2$
**7**         Obtain an estimate $\tilde{g}_{\hat{\mathbf{e}}}(\mathbf{y}_m)$ of $\nabla f(\mathbf{y}_m)^\top \hat{\mathbf{e}}$ with error at most $\epsilon'/(4R)$
**8**         **if** $|\tilde{g}_{\hat{\mathbf{e}}}(\mathbf{y}_m)| \leq \epsilon'/(4R)$ **then return** $\hat{\mathbf{y}} \leftarrow \mathbf{y}_m$
**9**         **else**
**10**           **if** $\tilde{g}_{\hat{\mathbf{e}}}(\mathbf{y}_m) > 0$ **then** $\mathbf{y}_r \leftarrow \mathbf{y}_m$ **else** $\mathbf{y}_l \leftarrow \mathbf{y}_m$
**11**     **until** $\|\mathbf{y}_r - \mathbf{y}_l\| \leq \epsilon'/L$
**12**     **return** $\hat{\mathbf{y}} \leftarrow \mathbf{y}_l$

---

*Proof.* Observe that in each iteration the value $\|\mathbf{y}_r - \mathbf{y}_l\|$ is decreased by at least $1/2$. Hence, the total number of iterations is at most

$$\log_2 \left( \frac{\|\mathbf{y}_{r0} - \mathbf{y}_{l0}\|}{\epsilon'/L} \right) \leq \log_2 \left( \frac{2RL}{\epsilon'} \right).$$

Within each iteration, we need to estimate the value $\nabla f(\mathbf{y}_m)^\top \hat{\mathbf{e}}$, which is the component of $\nabla f(\mathbf{y}_m)$ along $\hat{\mathbf{e}}$. Note that this is essentially a univariate mean estimation problem, given that $\hat{\mathbf{e}}$ is fixed in this iteration we can define an 1-dimensional random variable $\nabla f(\mathbf{y}_m)^\top \hat{\mathbf{e}}$ whose variance is at most

$$\text{Var}[\nabla f(\mathbf{y}_m)^\top \hat{\mathbf{e}}] \leq \mathbb{E}|\nabla f(\mathbf{y}_m)^\top \hat{\mathbf{e}}|^2 \leq \mathbb{E}\|\nabla f(\mathbf{y}_m)\|^2 \leq L^2.$$

Then by Lemma 7, with success probability at least

$$1 - \left( 6\mathcal{T} \log_2 \left( \frac{2RL}{\epsilon'} \right) \right)^{-1}, \tag{15}$$

an estimate of $\nabla f(\mathbf{y}_m)^\top \hat{\mathbf{e}}$ with error at most $\epsilon'/(4R)$ can be obtained using

$$\frac{L}{\epsilon'/(4R)} \cdot \log \left( 6\mathcal{T} \log_2 \left( \frac{2RL}{\epsilon'} \right) \right) = \widetilde{\mathcal{O}} \left( \frac{RL}{\epsilon'} \right)$$

queries to the quantum stochastic gradient oracle $O_{\mathbf{g}}$, given that we can prepare the following quantum oracle

$$O_{\mathbf{g}^\top \hat{\mathbf{e}}} |\mathbf{x}\rangle \otimes |0\rangle \rightarrow |\mathbf{x}\rangle \otimes \int_{\mathbf{v} \in \mathbb{R}^d} \sqrt{p_{f,\mathbf{x}}(\mathbf{v})\mathrm{d}\mathbf{v}} \, |\mathbf{v}^\top \hat{\mathbf{e}}\rangle \otimes |\text{garbage}(\mathbf{v})\rangle,$$

using one query to $O_{\mathbf{g}}$. Hence, Algorithm 4 uses $\widetilde{\mathcal{O}}(RL/\epsilon')$ queries to the quantum stochastic gradient oracle $O_{\mathbf{g}}$, with failure probability at most

$$\log_2 \left( \frac{\|\mathbf{y}_{r0} - \mathbf{y}_{l0}\|}{\epsilon'/L} \right) \cdot \left( 6\mathcal{T} \log_2 \left( \frac{2RL}{\epsilon'} \right) \right)^{-1} \leq \frac{1}{6\mathcal{T}} \tag{16}$$

by union bound. Next, we show that the output $\hat{\mathbf{y}}$ of Algorithm 4 has a relatively small function value as desired. We denote

$$\lambda^* := \underset{\lambda \in [0,1]}{\arg\min} f(\lambda \mathbf{y}_{l0} + (1-\lambda)\mathbf{y}_{r0}) \quad \text{and} \quad \mathbf{y}^* := \lambda^* \mathbf{y}_{l0} + (1-\lambda^*)\mathbf{y}_{r0}.$$

If the algorithm terminates at a point $\hat{\mathbf{y}}$ satisfying $|\tilde{g}_{\hat{\mathbf{e}}}(\hat{\mathbf{y}})| \leq \epsilon'/(4R)$, by convexity and the Cauchy Schwarz inequality we have

$$f(\hat{\mathbf{y}}) - f(\mathbf{y}^*) \leq |\nabla f(\hat{\mathbf{y}})^\top \hat{\mathbf{e}}| \cdot \|\hat{\mathbf{y}} - \mathbf{y}^*\| \leq \frac{\epsilon'}{2R} \cdot 2R \leq \epsilon'.$$

Otherwise, by induction we can demonstrate that in each iteration of the algorithm, $\mathbf{y}^*$ resides within the segment bounded by $\mathbf{y}_l$ and $\mathbf{y}_r$. Assume that this assertion holds true for the $t$-th iteration. If $\tilde{g}_{\hat{\mathbf{e}}}(\mathbf{y}_m) > \epsilon'/(4R)$, we have

$$\langle \nabla f(\mathbf{y}_m), \mathbf{y}_l - \mathbf{y}_m \rangle = -\langle \nabla f(\mathbf{y}_m), \hat{\mathbf{e}} \rangle \cdot \|\mathbf{y}_l - \mathbf{y}_m\|$$
$$\leq - \left( \tilde{g}_{\hat{e}}(\mathbf{y}_m) - \frac{\epsilon'}{4R} \right) \|\mathbf{y}_l - \mathbf{y}_m\| < 0,$$

indicating that the value of $f$ will decrease along the direction $\mathbf{y}_l - \mathbf{y}_m$. Hence, $\mathbf{y}^*$ lies in the segment between $\mathbf{y}_l$ and $\mathbf{y}_m$ of the $t$-th iteration, or equivalently, the segment between $\mathbf{y}_l$ and $\mathbf{y}_m$ of the $(t+1)$-th iteration, given that $f$ is convex. A symmetric argument applies in the case of $\tilde{g}_{\hat{\mathbf{e}}}(\mathbf{y}_m) < \epsilon'/(4R)$. Then, we have

$$\|\mathbf{y}_l - \mathbf{y}^*\| \leq \|\mathbf{y}_l - \mathbf{y}_r\|$$

for every iteration, which leads to

$$\|\hat{\mathbf{y}} - \mathbf{y}^*\| \leq \frac{\epsilon'}{L}$$

when the algorithm terminates. Hence,

$$f(\hat{\mathbf{y}}) - f(\mathbf{y}^*) \leq L \cdot \|\hat{\mathbf{y}} - \mathbf{y}^*\| = \epsilon'$$

considering that $f$ is $L$-Lipschitz. $\qquad\square$

Using Algorithm 4 as a subroutine, we develop the following Algorithm 5 that solves Problem 5.

---

**Algorithm 5:** Stochastic Approximately best point among finite number of points

---

**Input:** A set of points $\{\mathbf{x}_1, \ldots, \mathbf{x}_{\mathcal{T}}\} \subset \mathbb{B}_R(\mathbf{0})$ where $\mathcal{T}$ is a power of 2, accuracy $\epsilon$
**Output:** $\hat{\mathbf{x}}$ such that $f(\hat{\mathbf{x}}) \leq \min_i f(\mathbf{x}_i) + \epsilon$.
1 $\mathbf{y}_{0,i} \leftarrow \mathbf{x}_i$ for all $i \in [\mathcal{T}]$
2 **for** $\tau = 1, \ldots, \log_2 \mathcal{T}$ **do**
3 $\quad$ **for** $j = 1, \ldots, \mathcal{T}/2^{\tau}$ **do** $\mathbf{y}_{\tau,j} =$ StochasticLineSearch$(\mathbf{y}_{\tau-1,2j-1}, \mathbf{y}_{\tau-1,2j}, \epsilon/\log_2 \mathcal{T})$
4 **return** $\mathbf{y}_{\log \mathcal{T}, 1}$

---

*Proof of Proposition 2.* We start by demonstrating that without loss of generality we can assume $\mathcal{T}$ to be a power of 2. If $\mathcal{T}$ is not already a power of 2, we can simply augment the set with at most $\mathcal{T}$ points, each being $\mathbf{x}_1$, without changing the algorithm's output.

Next, we show that Algorithm 5 makes $\mathcal{T} - 1$ calls to the StochasticLineSearch$(\cdot)$ subroutine (Algorithm 4) with an accuracy of $\epsilon' = \epsilon/\log_2 \mathcal{T}$ to solve the Problem 5 problem. The algorithm exhibits a hierarchical structure, as illustrated in Figure 1, where each non-leaf node invokes the StochasticLineSearch$(\cdot)$ subroutine once. Consequently, the total number of calls is $\mathcal{T} - 1$.

By Lemma 8, each call to StochasticLineSearch$(\cdot)$ is executed successfully with probability at least $1 - 1/(6\mathcal{T})$. Then by union bound, we can deduce that the likelihood of all calls to StochasticLineSearch$(\cdot)$ being successful is at least $5/6$. In this case, for any node $\mathbf{y}_{\tau,j}$ in the tree with $\tau \in [\log_2 \mathcal{T}]$ and $j \in [\mathcal{T}/2^{\tau}]$, by convexity and the guarantee of Lemma 8 we have

$$f(\mathbf{y}_{\tau,j}) \leq \min_{\lambda \in [0,1]} f\left( \lambda \mathbf{y}_{\tau-1,2j-1} + (1-\lambda)\mathbf{y}_{\tau-1,2j} \right) + \epsilon/\log_2 \mathcal{T}$$
$$\leq \min\{f(\mathbf{y}_{\tau-1,2j-1}), f(\mathbf{y}_{\tau-1,2j})\} + \epsilon/\log_2 \mathcal{T}.$$

Therefore for any leaf node $\mathbf{x}_j$ with $j \in [\mathcal{T}]$, by summing over the path between the root $\mathbf{y}_{t,1}$ and $\mathbf{x}_j$ we have

$$f(\mathbf{y}_{t,1}) \leq f(\mathbf{x}_j) + t \cdot \frac{\epsilon}{\log_2 \mathcal{T}} = f(\mathbf{x}_j) + \epsilon,$$

which leads to

$$f(\mathbf{y}_{t,1}) \leq \min_{j \in [\mathcal{T}]} f(\mathbf{x}_j) + \epsilon,$$

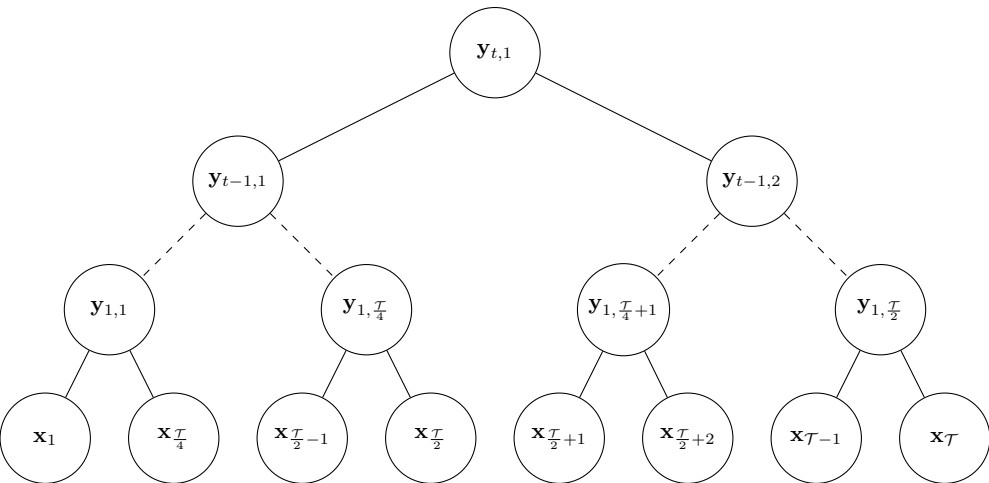

Figure 1: The hierarchical structure of Algorithm 5.

indicating $\mathbf{y}_{t,1}$ is a valid solution of Problem 5. Note that there are in total $\mathcal{T} - 1$ non-leaf nodes in Figure 1, and the value of each non-leaf node is computed by one call to the subroutine StochasticLineSearch$(\cdot)$. Hence, the total number of calls to StochasticLineSearch$(\cdot)$ equals $\mathcal{T} - 1$, and the overall failure probability of Algorithm 5 is at most

$$(\mathcal{T} - 1) \cdot \frac{1}{6\mathcal{T}} \leq \frac{1}{6} \tag{17}$$

by union bound, as the failure probability of Algorithm 4 is at most $1/(6\mathcal{T})$. Since each call to Algorithm 4 takes

$$\widetilde{\mathcal{O}}\left(\frac{RL}{\epsilon'}\right) = \widetilde{\mathcal{O}}\left(\frac{RL}{\epsilon}\right)$$

queries to the quantum stochastic gradient oracle $O_{\mathbf{g}}$ defined in (2), the total number of queries to the quantum stochastic gradient oracle $O_{\mathbf{g}}$ is then $\widetilde{\mathcal{O}}\left(RL\mathcal{T}/\epsilon\right)$. $\qquad\square$

### D.3 Proof of Corollary 1

*Proof of Corollary 1.* We first run cutting plane method with the separation oracle being an $\frac{\epsilon}{10R}$-approximate gradient of $f$ implemented by Algorithm 1, which by Proposition 1 outputs a set $\Gamma$ of $\mathcal{T} = \widetilde{\mathcal{O}}(d\log(L/\epsilon))$ points containing at least one $\mathcal{O}(\epsilon)$-optimal point of $f$ using $\widetilde{\mathcal{O}}\left(d^{3/2}LR/\epsilon\right)$ queries to $O_{\mathbf{g}}$ with success probability at least $5/6$. Then by running Algorithm 5 on $\Gamma$, with success probability at least $5/6$ we can find a point $\hat{\mathbf{x}} \in \mathbb{R}^d$ satisfying

$$f(\hat{\mathbf{x}}) \leq \min_{\mathbf{x} \in \Gamma} f(\mathbf{x}) + \epsilon \leq f(\mathbf{x}^*) + \mathcal{O}(\epsilon),$$

which takes $\widetilde{\mathcal{O}}(RL\mathcal{T}/\epsilon) = \widetilde{\mathcal{O}}(dRL/\epsilon)$ queries to $O_{\mathbf{g}}$ by Proposition 2. Hence, the overall number of queries to $O_{\mathbf{g}}$ equals $\widetilde{\mathcal{O}}(d^{3/2}LR/\epsilon)$, and the overall success probability is at least $2/3$. $\qquad\square$

## E Proof of Theorem 6

The proof of Theorem 6 is based on the following result from [40].

**Theorem 9** ([40, Theorem 1]). *Consider the bounded variance setting in Problem 2, the output $\mathbf{x}_{\mathrm{out}}$ of Algorithm 6 is an expected $\frac{2\Delta\ell}{\mathcal{T}} + \hat{\sigma}^2$-critical point.*

*Proof of Theorem 6.* By Theorem 9 and the associated parameter setting, the output $\mathbf{x}_{\mathrm{out}}$ of Algorithm 6 satisfies

$$\mathbb{E}\|\nabla f(\mathbf{x}_{\mathrm{out}})\|^2 \leq \frac{2\Delta\ell}{\mathcal{T}} + \hat{\sigma}^2 \leq \epsilon^2,$$

---
**Algorithm 6:** Quantum randomized SGD (Q-SGD)
---
**Input:** Function $f \colon \mathbb{R}^d \to \mathbb{R}$, precision $\epsilon$, variance $\sigma$, smoothness $\ell$
**Parameters:** $\hat{\sigma} = \epsilon/3$, total iteration budget $\mathcal{T} = 12\Delta\ell\epsilon^{-2}$
**Output:** $\epsilon$-critical point of $f$
1 Uniformly randomly select $N$ from $1, \ldots, \mathcal{T}$
2 Set $\mathbf{x}_0 \leftarrow \mathbf{0}$
3 **for** $t = 0, 1, 2, \ldots, N-1$ **do**
4     Call Algorithm 1 for an unbiased estimate $\tilde{\mathbf{g}}_t$ of $\nabla f(\mathbf{x}_t)$ with variance at most $\hat{\sigma}^2$
5     $\mathbf{x}_{t+1} \leftarrow \mathbf{x}_t - \tilde{\mathbf{g}}_t/\ell$
6 **return** $\mathbf{x}_N$
---

implying that Algorithm 6 solves Problem 2 with probability at least $2/3$ under the bounded variance setting, and the remaining thing would be to analyze the number of queries it makes to $O_{\mathbf{g}}$. By Theorem 4, at each iteration one can run Algorithm 1 to obtain an unbiased estimate $\tilde{\mathbf{g}}_t$ of $\nabla f(\mathbf{x}_t)$ with variance at most $\hat{\sigma}^2$ using

$$\widetilde{\mathcal{O}}\left(\frac{\sigma\sqrt{d}}{\hat{\sigma}}\right) = \widetilde{\mathcal{O}}\left(\frac{\sigma\sqrt{d}}{\epsilon}\right)$$

queries to $O_{\mathbf{g}}$. Then, the total number of queries to $O_{\mathbf{g}}$ is at most

$$\mathcal{T} \cdot \widetilde{\mathcal{O}}\left(\frac{\sigma\sqrt{d}}{\epsilon}\right) = \widetilde{\mathcal{O}}\left(\frac{\sigma\Delta\ell\sqrt{d}}{\epsilon^3}\right).$$

$\square$

## F   Proof of Theorem 7

---
**Algorithm 7:** Q-SPIDER
---
**Input:** Function $f \colon \mathbb{R}^d \to \mathbb{R}$, precision $\epsilon$, variance $\sigma$, smoothness $\ell$
**Parameters:** $q = \frac{20\sigma}{\epsilon}$, $\hat{\sigma}_1 = \frac{\epsilon}{40}$, $\hat{\sigma}_2 = \frac{\epsilon}{40}\sqrt{\frac{\epsilon}{10\sigma}}$, total iteration budget $\mathcal{T} = \frac{1600\ell\Delta}{\sigma^2}$
**Output:** An $\epsilon$-critical point of $f$
1 Set $\mathbf{x}_0 \leftarrow \mathbf{0}$
2 **for** $t = 0, 1, 2, \ldots, \mathcal{T}$ **do**
3     **if** $\mod(t, q) = 0$ **then**
4        Call Algorithm 1 to obtain an unbiased estimate $\tilde{\mathbf{g}}_t$ of $\nabla f(\mathbf{x}_t)$ with variance $\hat{\sigma}_1^2$
5        $\mathbf{v}_t \leftarrow \tilde{\mathbf{g}}_t$
6     **else**
7        Call Algorithm 1 to obtain an unbiased estimate $\tilde{\mathbf{g}}_t$ of $\nabla f(\mathbf{x}_t) - \nabla f(\mathbf{x}_{t-1})$ with variance $\hat{\sigma}_2^2$
8        $\mathbf{v}_t \leftarrow \tilde{\mathbf{g}}_t + \mathbf{v}_{t-1}$
9     **if** $\|\mathbf{v}_t\| \leq 2\epsilon$ **then return** $\mathbf{x}_t$;
10     **else** $\mathbf{x}_{t+1} \leftarrow \mathbf{x}_t - \frac{\epsilon}{\ell} \cdot \frac{\mathbf{v}_t}{\|\mathbf{v}_t\|}$
11 **return** $\mathbf{x}_{\mathcal{T}}$
---

The proof of Theorem 7 is based on the following result from [29].

**Theorem 10** ([29, Theorem 1])**.** *Consider the mean-squared smoothness setting in Problem 2, the output $\mathbf{x}_{\text{out}}$ of Algorithm 7 is an expected $\epsilon$-critical point.*

*Proof of Theorem 7.* Theorem 10 shows that Algorithm 7 can solve Problem 2 under the mean-squared smoothness setting and the remaining thing would be to analyze the number of queries it makes to $O_{\mathbf{g}}^S$.

For each iteration $t$ with $\mathrm{mod}(t,q) = 0$ such that Line 5 is executed, note that we can prepare the following quantum oracle

$$\hat{O}_{\mathbf{g}}\,|\mathbf{x}\rangle \otimes |0\rangle \to |\mathbf{x}\rangle \otimes \int_\omega \sqrt{p(\omega)\mathrm{d}\omega}\,|\mathbf{g}(\mathbf{x},\omega)\rangle \otimes |\mathrm{garbage}(\mathbf{x},\omega)\rangle,$$

where $p(\omega)$ is the probability distribution of the random seed $\omega$, by first applying $O_{\mathbf{g}}^S$ to the state

$$|\mathbf{x}\rangle \otimes \left(\int_\omega \sqrt{p(\omega)\mathrm{d}\omega}\,|\omega\rangle\right) \otimes |0\rangle$$

and then uncompute the second register. This procedure uses one query to the $\sigma$-SQ-QSGO $O_{\mathbf{g}}^S$ (Definition 4). By Theorem 4, one can run Algorithm 1 to obtain an unbiased estimate $\tilde{\mathbf{g}}_t$ of $\nabla f(\mathbf{x}_t)$ with variance at most $\hat{\sigma}_1^2$ using

$$\widetilde{\mathcal{O}}\left(\frac{\sigma\sqrt{d}}{\hat{\sigma}_1}\right) = \widetilde{\mathcal{O}}\left(\frac{\sigma\sqrt{d}}{\epsilon}\right)$$

queries to $\hat{O}_{\mathbf{g}}$ and thus the same number of queries to $O_{\mathbf{g}}^S$.

Similarly, for each iteration $t$ with $\mathrm{mod}(t,q) = 0$ such that Line 7 is executed, note that one can prepare the following quantum oracle

$$\hat{O}_{\mathbf{g}}^t\,|\mathbf{x}_t\rangle \otimes |\mathbf{x}_{t-1}\rangle \otimes |0\rangle \to |\mathbf{x}_t\rangle \otimes |\mathbf{x}_{t-1}\rangle$$
$$\otimes \int_\omega \sqrt{p(\omega)\mathrm{d}\omega}\,|\mathbf{g}(\mathbf{x}_t,\omega) - \mathbf{g}(\mathbf{x}_{t-1},\omega)\rangle \otimes |\mathrm{garbage}(\mathbf{x}_t,\mathbf{x}_{t-1},\omega)\rangle$$

by first applying $O_{\mathbf{g}}^S$ twice to obtain the state

$$|\mathbf{x}\rangle \otimes |\mathbf{x}_{t-1}\rangle \otimes \int_\omega \sqrt{p(\omega)\mathrm{d}\omega}\,|\omega\rangle \otimes |\mathbf{g}(\mathbf{x},\omega)\rangle \otimes |\mathbf{g}(\mathbf{x}_{t-1},\omega)\rangle$$
$$\otimes |\mathbf{g}(\mathbf{x},\omega) - \mathbf{g}(\mathbf{x}_{t-1},\omega)\rangle \otimes |\mathrm{garbage}(\mathbf{x}_t,\mathbf{x}_{t-1},\omega)\rangle$$

and then uncompute the fourth and the fifth register. Observe that

$$\mathbb{E}_\omega\|\mathbf{g}(\mathbf{x}_t,\omega) - \mathbf{g}(\mathbf{x}_{t-1},\omega)\|^2 \le \ell^2\|\mathbf{x}_t - \mathbf{x}_{t-1}\| \le \epsilon^2,$$

then by Theorem 4, one can run Algorithm 1 to obtain an unbiased estimate $\tilde{\mathbf{g}}_t$ of $\nabla f(\mathbf{x}_t) - \nabla f(\mathbf{x}_{t-1})$ with variance at most $\hat{\sigma}_1^2$ using

$$\widetilde{\mathcal{O}}\left(\frac{\epsilon\sqrt{d}}{\hat{\sigma}_2}\right) = \widetilde{\mathcal{O}}\left(\sqrt{\frac{d\sigma}{\epsilon}}\right)$$

queries to $\hat{O}_{\mathbf{g}}^t$ and thus twice the number of queries to $O_{\mathbf{g}}^S$. Then, the total number of queries to $O_{\mathbf{g}}^S$ can be expressed as

$$\frac{\mathcal{T}}{q}\cdot\widetilde{\mathcal{O}}\left(\frac{\sigma\sqrt{d}}{\epsilon}\right) + \mathcal{T}\cdot\widetilde{\mathcal{O}}\left(\sqrt{\frac{d\sigma}{\epsilon}}\right) = \widetilde{\mathcal{O}}\left(\frac{\ell\Delta\sqrt{d}}{\epsilon^2}\left(1 + \sqrt{\frac{\sigma}{\epsilon}}\right)\right).$$

$\square$

# G  Proof of lower bounds

## G.1  Proof of Proposition 3

Note that Problem 3 is strictly harder than the multivariate mean estimation problem considered in [25]. In the latter, the objective is to produce an estimate of the expected value within a bounded $\ell_2$ error, without the requirement of unbiasedness. A query lower bound for solving quantum mean estimation was established in [25].

**Lemma 9** ([25], Theorem 3.8). *Let $\sigma \geq 0$ and denote $P_\sigma$ to be the set of all $d$-dimensional quantum random variables with covariance matrix $\Sigma$ such that $\mathrm{Tr}(\Sigma) = \sigma^2$. Suppose every $X \in P_\sigma$ is indexed by some random seeds $\omega$, i.e.,*

$$X = X(\omega), \quad \omega \sim p_\Omega,$$

*where $p_\Omega$ denotes the probability distribution of the random seed $\omega$. Then there exists a constant $\alpha$ such that for any $n > \alpha d$ and any quantum algorithm that uses at most $n$ queries to the following two oracles*

$$O_\Omega |0\rangle \to \int_\omega \sqrt{p_\Omega(\omega)\mathrm{d}\omega} \, |\omega\rangle \qquad and \qquad \mathcal{B}_X |\omega\rangle |0\rangle \to |\omega\rangle |X(\omega)\rangle \,,$$

*there exists an instance $X \in P_\sigma$ such that the quantum algorithm returns a mean estimate $\tilde{\mu}$ of the mean $\mu$ of $X$ that satisfies*

$$\|\tilde{\mu} - \mu\| \geq \Omega\left(\frac{\sigma\sqrt{d}}{n}\right)$$

*with probability at least $2/3$.*

By employing an oracle reduction argument, we obtain our lower bound result based on Lemma 9.

*Proof of Proposition 3.* We first show that one query to the quantum sampling oracle $O_X$ defined in Definition 1 can be implemented using one query to $O_\Omega$ and one query to $\mathcal{B}_X$. In particular, observe that

$$\mathcal{B}_X O_\Omega |0\rangle \otimes |0\rangle = \int_\omega \sqrt{p_\Omega(\omega)\mathrm{d}\omega} \, |\omega\rangle \otimes |X(\omega)\rangle \,.$$

By measuring the first register in the computational basis spanned by all the random seeds $\{|\omega\rangle\}$, we can obtain the desired output of $O_X$. Hence, if there exists a quantum algorithm making at most $\mathcal{O}(L\sqrt{d}/\hat{\sigma})$ queries to the oracle $O_X$ that solves Problem 3 with success probability at least $2/3$, there also exists a quantum mean estimation algorithm that uses at most $\mathcal{O}(L\sqrt{d}/\hat{\sigma})$ queries to $O_\Omega$ and $\mathcal{B}_X$ and for any $X \in P_\sigma$ it outputs an estimate $\tilde{\mu}$ of the mean $\mu$ of $X$ that satisfies $\|\tilde{\mu} - \mu\| \leq \mathcal{O}(\hat{\sigma})$ with probability at least $2/3$, which contradicts to Lemma 9. $\square$

### G.2 Proof of Theorem 8

Our proof follow a similar approach to the previous proof of the quantum mean estimation lower bound [25]. Specifically, our results are derived through a reduction to a specific multivariate mean estimation problem. In this problem, all vectors are sampled from a set of orthonormal bases and there might be duplicated vectors. We choose to reduce to this particular problem instead of directly reducing to mean estimation problem because the construction of our hard instances require a tighter control over the norm of the vectors as well as the norm of their expected value. This problem can be equivalently represented as the following composition problem, as defined in [25].

**Problem 6** (Search$^N \circ$ Parity$^M$). *Let $N, M \geq 1$ be two integers. Let $\mathcal{A}_{N,M}$ denote the set of matrices $A \in \{0,1\}^{N \times M}$ such that $\lfloor N/2 \rfloor$ rows have Hamming weights $\lfloor M/2 \rfloor$, and the other rows have Hamming weights $\lfloor M/2 \rfloor + 1$. Define the vector $\mathbf{b}^{(A)}$ such that,*

$$b_i^{(A)} = \begin{cases} 0, & \text{if the $i$-th row of $A$ has Hamming weight } \lfloor M/2 \rfloor, \\ 1, & \text{if the $i$-th row of $A$ has Hamming weight } \lfloor M/2 \rfloor + 1, \end{cases} \tag{18}$$

*for each $i \in [N]$. Then, the Search$^N \circ$ Parity$^M$ problem consists of finding a vector $\widetilde{\mathbf{b}} \in \mathbb{R}^N$ that minimizes $\|\widetilde{\mathbf{b}} - \mathbf{b}^{(A)}\|_1$ given a quantum oracle*

$$O_A |i, j\rangle \to (-1)^{A_{ij}} |i, j\rangle \,. \tag{19}$$

[25] showed that $\Omega(NM)$ queries are needed to approximate the vector $\mathbf{b}^{(A)}$ with small error.

**Lemma 10** ([25, Lemma 3.6]). *There exists a constant $\alpha > 1$ such that, for any quantum algorithm for the $\mathsf{Search}^N \circ \mathsf{Parity}^M$ problem that uses at most $NM/\alpha$ queries there exists an $A \in \mathcal{A}_{N,M}$ such that this algorithm returns a vector $\widetilde{\mathbf{b}}$ satisfying $\|\widetilde{\mathbf{b}} - \mathbf{b}^{(A)}\| \geq \sqrt{N}/2$ with probability at least $2/3$.*

We establish our quantum lower bounds for stochastic convex optimization by establishing a correspondence between the $\mathsf{Search}^N \circ \mathsf{Parity}^M$ problem and Problem 1. Specifically, for any input $A \in \mathcal{A}_{N \times M}$ to the $\mathsf{Search}^N \circ \mathsf{Parity}^M$ problem, we design the following convex function whose optimal point corresponds to the solution of the $\mathsf{Search}^N \circ \mathsf{Parity}^M$ problem under certain sets of parameters,

$$\bar{f}^A(\mathbf{x}) := \mathbb{E}_{i,j}[f_{i,j}(\mathbf{x})], \tag{20}$$

where

$$f_{i,j}^A(\mathbf{x}) := -\frac{1}{3} \langle \mathbf{x}, \mathbf{g}_{i,j} \rangle + \frac{2L}{3} \cdot \max\left\{0, \|\mathbf{x}\| - \frac{R}{2}\right\}$$

for some $\mathbf{g}_{i,j} \in \mathbb{R}^d$ satisfying $\|\mathbf{g}_{i,j}\| \leq L$ for all $i \in [N]$ and $j \in [M]$.

**Lemma 11.** *Denote $\bar{\mathbf{g}} := \mathbb{E}_{i,j}[\mathbf{g}_{i,j}]$. Then, the function $\bar{f}^A$ defined in (20) has the following properties if $\bar{\mathbf{g}}$ is a non-zero vector.*

1. *$\bar{f}^A$ is convex.*

2. *$\bar{f}^A$ is minimized at $\mathbf{x}^* = \frac{R}{2} \frac{\bar{\mathbf{g}}}{\|\bar{\mathbf{g}}\|}$.*

3. *Every $\epsilon$-optimum $\mathbf{x}$ of $\bar{f}^A$ satisfies*

$$\left\langle \frac{R}{2} \cdot \frac{\mathbf{x}}{\|\mathbf{x}\|}, \mathbf{x}^* \right\rangle \geq 1 - \frac{R\epsilon}{2\|\bar{\mathbf{g}}\|}$$

4. *For*

$$\hat{\mathbf{g}}_{i,j}(\mathbf{x}) := -\frac{1}{3} \cdot \mathbf{g}_{i,j} + \frac{2L}{3} \cdot \max\left\{\|\mathbf{x}\| - \frac{R}{2}, 0\right\} \cdot \frac{\mathbf{x}}{\|\mathbf{x}\|} \tag{21}$$

   *it is the case that*

$$\mathbb{E}_{i,j}[\hat{\mathbf{g}}_{i,j}(\mathbf{x})] \in \partial \bar{f}^A(\mathbf{x}) \quad and \quad \|\hat{\mathbf{g}}_{i,j}(\mathbf{x})\| \leq L, \qquad \forall \mathbf{x} \in \mathbb{R}^d.$$

*Proof.* Since $\bar{f}^A$ is the sum of a linear function and a maximum function, both of which are convex, $\bar{f}^A$ itself is convex.

Observe that the minimum of $\bar{f}^A$ cannot be achieved when $\|\mathbf{x}\| > R/2$ for otherwise the function value can be further decreased when moving towards $\mathbf{0}$ given that

$$\|\bar{\mathbf{g}}\| = \left\|\frac{\sum_{i,j} \mathbf{g}_{i,j}}{NM}\right\| \leq \frac{\sum_{i,j} \|\mathbf{g}_{i,j}\|}{NM} \leq L.$$

When $\|\mathbf{x}\| \leq R/2$, $\bar{f}^A$ can be expressed as

$$\bar{f}^A(\mathbf{x}) = -\frac{1}{3} \langle \mathbf{x}, \bar{\mathbf{g}} \rangle,$$

which is minimized at

$$\frac{R}{2} \cdot \frac{\bar{\mathbf{g}}}{\|\bar{\mathbf{g}}\|} = \frac{R\mathbf{b}^{(A)}}{\sqrt{2d}}.$$

For any $\mathbf{x}$ that is an $\epsilon$-optimum of $\bar{f}^A$, we define

$$\mathbf{x}' = \begin{cases} \mathbf{x}, & \|\mathbf{x}\| \leq \frac{R}{2}, \\ \frac{R}{2} \cdot \frac{\mathbf{x}}{\|\mathbf{x}\|}, & \text{otherwise,} \end{cases}$$

which satisfies $f(\mathbf{x}') \le f(\mathbf{x})$ and is thus also an $\epsilon$-optimum of $f$. Then we can derive that

$$-\frac{1}{3}\langle \mathbf{x}', \bar{\mathbf{g}}\rangle + \frac{1}{3}\langle \mathbf{x}^*, \bar{\mathbf{g}}\rangle \le \epsilon,$$

and

$$\langle \mathbf{x}', \mathbf{x}^*\rangle \ge \langle \mathbf{x}^*, \mathbf{x}^*\rangle - \epsilon \cdot \frac{\|\mathbf{x}^*\|}{\|\bar{\mathbf{g}}\|}$$

$$\ge 1 - \frac{R\epsilon}{2\|\bar{\mathbf{g}}\|},$$

which leads to

$$\left\langle \frac{R}{2}\cdot\frac{\mathbf{x}}{\|\mathbf{x}\|}, \mathbf{x}^*\right\rangle \ge 1 - \frac{R\epsilon}{2\|\bar{\mathbf{g}}\|}$$

given that $\frac{R}{2}\frac{\mathbf{x}}{\|\mathbf{x}\|} \ge \|\mathbf{x}'\|$. As for the last entry, note that

$$\mathbb{E}_{i,j}[\hat{\mathbf{g}}_{i,j}(\mathbf{x})] = -\frac{1}{3}\cdot\mathbb{E}_{i,j}\mathbf{g}_{i,j} + \frac{2L}{3}\cdot\max\left\{\|\mathbf{x}\| - \frac{R}{2}\right\}\cdot\frac{\mathbf{x}}{\|\mathbf{x}\|}$$

$$= -\frac{1}{3}\cdot\bar{\mathbf{g}} + \frac{2L}{3}\cdot\max\left\{\|\mathbf{x}\| - \frac{R}{2}\right\}\cdot\frac{\mathbf{x}}{\|\mathbf{x}\|},$$

which is the (sub-)gradient of $\bar{f}^A$ at $\mathbf{x}$. Moreover, for any $i, j$ we have

$$\|\hat{\mathbf{g}}_{i,j}(\mathbf{x})\| \le \frac{1}{3}\|\mathbf{g}_{i,j}\| + \frac{2L}{3} \le L.$$

$\square$

Equipped with Lemma 11, we first prove our quantum lower bound for Problem 1 in the low-dimensional regime by encoding the Search $\circ$ Parity problem into the task of finding an $\epsilon$-optimal point of $\bar{f}^A$ defined in (20).

*Proof of Theorem 8 when $\epsilon \le \frac{RL}{100\sqrt{d}}$.* Our proof proceeds by contradiction. Denote $n = \frac{RL\sqrt{d}}{100\alpha\epsilon}$. Assume for simplicity that $d$ is even, and $\alpha n$ is an even multiple of $d$ (the other cases can be handled by padding arguments). Then, we show that $\mathsf{Search}^d \circ \mathsf{Parity}^{\alpha n/d}$ on this instance $A$ can be solved by finding an $\epsilon$-optimum of $\bar{f}^A$ defined in (20). In particular, we define the set of vectors $\{\mathbf{g}_{i,j}\}$ to be

$$\mathbf{g}_{i,j} = \frac{\alpha L n}{\sqrt{(2\alpha n)^2 - d^2}}(-1)^{1+A_{i,j}}\mathbf{e}_i, \tag{22}$$

where $i \in [d]$, $j \in [\alpha n/d]$, and $\mathbf{e}_i \in \mathbb{R}^d$ is the $i$-th indicator vector. Consider the function $\bar{f}^A$ defined in (20) with the set of vectors $\{\mathbf{g}_{i,j}\}$ having the values in (22), we have

$$\bar{\mathbf{g}} = \mathbb{E}_{i,j}[\mathbf{g}_{i,j}] = \frac{1}{\alpha n}\cdot\frac{\alpha L n}{\sqrt{(2\alpha n)^2 - d^2}}\sum_{i=1}^{d}\mathbf{e}_i\sum_{j=1}^{\alpha n/d}(-1)^{1+A_{i,j}}$$

$$= \frac{2L}{\sqrt{(2\alpha n)^2 - d^2}}\sum_{i=1}^{d}b_i^{(A)} = \frac{2L}{\sqrt{(2\alpha n)^2 - d^2}}\mathbf{b}^{(A)},$$

where the vector $\mathbf{b}^{(A)}$ is defined in (18). By Lemma 11, every $\epsilon$-optimal point $\mathbf{x}$ of $\bar{f}^A$ satisfies

$$\left\langle \frac{R}{2}\frac{\mathbf{x}}{\|\mathbf{x}\|}, \mathbf{x}^*\right\rangle \ge 1 - \frac{R\epsilon}{2\|\bar{\mathbf{g}}\|},$$

by which we can derive that

$$\left\langle \frac{R}{2}\frac{\mathbf{x}}{\|\mathbf{x}\|}, \mathbf{b}^{(A)}\right\rangle \ge \frac{\|\mathbf{b}^{(A)}\|}{\|\mathbf{x}^*\|}\cdot\left(1 - \frac{R\epsilon}{2\|\bar{\mathbf{g}}\|}\right)$$

$$\ge \frac{R}{2}\sqrt{\frac{d}{2}} - \frac{3\alpha n\epsilon}{L} = \frac{R\sqrt{d}}{4},$$

which leads to

$$\left\langle \sqrt{\frac{d}{2}} \cdot \frac{\mathbf{x}}{\|\mathbf{x}\|}, \mathbf{b}^{(A)} \right\rangle \geq \frac{d}{2\sqrt{2}}.$$

Given that

$$\left\| \sqrt{\frac{d}{2}} \cdot \frac{\mathbf{x}}{\|\mathbf{x}\|} \right\| = \left\| \mathbf{b}^{(A)} \right\| = \sqrt{\frac{d}{2}},$$

we have

$$\left\| \sqrt{\frac{d}{2}} \cdot \frac{\mathbf{x}}{\|\mathbf{x}\|} - \mathbf{b}^{(A)} \right\| \leq \frac{\sqrt{d}}{2},$$

indicating that we can obtain an estimate $\widetilde{\mathbf{b}} = \sqrt{\frac{d}{2}} \frac{\mathbf{x}}{\|\mathbf{x}\|}$ of $\mathbf{b}^{(A)}$ with $\ell_2$-error at most $\sqrt{d}/2$ by obtaining an $\epsilon$-optimal point of $\bar{f}^A$.

Next, we show that we can implement a quantum stochastic gradient oracle of $\bar{f}^A$ using only one query to the oracle $O_A$, given that one can use one query to $O_A$ to implement the following oracle

$$O_{\mathbf{g}}^A \, |i\rangle \, |j\rangle \, |0\rangle \to |i\rangle \, |j\rangle \, |\mathbf{g}_{i,j}\rangle$$

and use one query to $O_{\mathbf{g}}^A$ to implement

$$O_{\hat{\mathbf{g}}}^A \, |\mathbf{x}\rangle \, |i\rangle \, |j\rangle \, |0\rangle \to |\mathbf{x}\rangle \, |i\rangle \, |j\rangle \, |\hat{\mathbf{g}}_{i,j}(\mathbf{x})\rangle,$$

where $\hat{\mathbf{g}}_{i,j}(\mathbf{x})$ is defined in (21). Then by applying $O_{\hat{\mathbf{g}}}^A$ to the state

$$\frac{1}{\sqrt{\alpha n}} \, |\mathbf{x}\rangle \sum_{i \in [d]} \sum_{j \in [\alpha n/d]} |i\rangle \, |j\rangle \, |0\rangle$$

and uncompute the second and the third register, we construct the following stochastic gradient oracle $O_{\hat{\mathbf{g}}}$ of $\bar{f}^A$.

$$O_{\hat{\mathbf{g}}} \, |\mathbf{x}\rangle \otimes |0\rangle \to \frac{1}{\sqrt{\alpha n}} \, |\mathbf{x}\rangle \otimes \sum_{i \in [d]} \sum_{j \in [\alpha n/d]} |\hat{\mathbf{g}}_{i,j}(\mathbf{x})\rangle.$$

Hence, any quantum algorithm that can find an $\epsilon$-optimum of $\bar{f}^A$ using $T$ queries to $O_{\hat{\mathbf{g}}}$ can be transformed into a quantum algorithm for the $\mathsf{Search}^d \circ \mathsf{Parity}^{\alpha n/d}$ problem that uses $T$ queries to the oracle $O_A$ defined in (19) and returns an estimate $\widetilde{\mathbf{b}}$ with $\ell_2$-error at most $\sqrt{d}/2$. Then by Lemma 10, for any quantum algorithm making less than

$$d \cdot \frac{\alpha n}{d} \cdot \frac{1}{\alpha} = \frac{RL}{2\epsilon\alpha} \sqrt{\frac{d}{2}}$$

queries to the quantum stochastic gradient oracle $O_{\hat{\mathbf{g}}}$, there exists an $A \in \mathcal{A}_{d,\alpha n/d}$ and corresponding $\bar{f}^A$ such that the output of the algorithm is not an $\epsilon$-optimum of $\bar{f}^A$ with probability at least $2/3$. $\quad\square$

Similarly, we can obtain our quantum lower bound in the high-dimensional regime by encoding the Search $\circ$ Parity problem into the task of finding an $\epsilon$-optimal point of $\bar{f}^A$ defined in (20).

*Proof of Theorem 8 when $\frac{RL}{100\sqrt{d}} \leq \epsilon \leq 1$.* Our proof proceeds by contradiction. Denote $n = \frac{R^2 L^2}{10000\alpha\epsilon^2}$. Consider an instance $A \in \mathcal{A}_{\alpha n, 1}$ of the $\mathsf{Search}^{\alpha n} \circ \mathsf{Parity}^1$ problem where $d$ is a power of 2 and $\alpha$ is even. Then, we show that $\mathsf{Search}^d \circ \mathsf{Parity}^{\alpha n/d}$ on this instance $A$ can be solved by finding an $\epsilon$-optimum of $\bar{f}^A$ defined in (20). In particular, we define the set of vectors $\{\mathbf{g}_{i,j}\}$ where $j \equiv 1$ to be

$$\mathbf{g}_{i,1} = \alpha L A_{i,1} \sqrt{\frac{n}{2\alpha^2 n - 2\alpha}} \mathbf{e}_i, \tag{23}$$

where $i \in [\alpha n]$ and $\mathbf{e}_i \in \mathbb{R}^d$ is the $i$-th indicator vector. Consider the function $\bar{f}^A$ defined in (20) with the set of vectors $\{\mathbf{g}_{i,1}\}$ having the values in (23), we have

$$\bar{\mathbf{g}} = \mathbb{E}_i[\mathbf{g}_{i,1}] = \frac{\alpha L}{\alpha n} \sqrt{\frac{n}{2\alpha^2 n - 2\alpha}} \sum_{i=1}^d A_{i,1} \mathbf{e}_i = \frac{L}{n} \sqrt{\frac{n}{2\alpha^2 n - 2\alpha}} \mathbf{b}^{(A)},$$

where the vector $\mathbf{b}^{(A)}$ is defined in (18). By Lemma 11, every $\epsilon$-optimal point $\mathbf{x}$ of $\bar{f}^A$ satisfies

$$\left\langle \frac{R}{2} \frac{\mathbf{x}}{\|\mathbf{x}\|}, \mathbf{x}^* \right\rangle \geq 1 - \frac{R\epsilon}{2\|\bar{\mathbf{g}}\|},$$

by which we can derive that

$$\left\langle \frac{R}{2} \frac{\mathbf{x}}{\|\mathbf{x}\|}, \mathbf{b}^{(A)} \right\rangle \geq \frac{\|\mathbf{b}^{(A)}\|}{\|\mathbf{x}^*\|} \cdot \left(1 - \frac{R\epsilon}{2\|\bar{\mathbf{g}}\|}\right)$$

$$\geq \frac{R}{2} \sqrt{\frac{\alpha n}{2}} - \frac{6\alpha n \epsilon}{L} = \frac{R\sqrt{\alpha n}}{4},$$

which leads to

$$\left\langle \sqrt{\frac{\alpha n}{2}} \cdot \frac{\mathbf{x}}{\|\mathbf{x}\|}, \mathbf{b}^{(A)} \right\rangle \geq \frac{\alpha n}{2\sqrt{2}}.$$

Given that

$$\left\| \sqrt{\frac{\alpha n}{2}} \cdot \frac{\mathbf{x}'}{\|\mathbf{x}'\|} \right\| = \left\| \mathbf{b}^{(A)} \right\| = \sqrt{\frac{\alpha n}{2}},$$

we have

$$\left\| \sqrt{\frac{\alpha n}{2}} \cdot \frac{\mathbf{x}}{\|\mathbf{x}\|} - \mathbf{b}^{(A)} \right\| \leq \frac{\sqrt{\alpha n}}{2}.$$

indicating that we can obtain an estimate $\widetilde{\mathbf{b}} = \sqrt{\frac{\alpha n}{2}} \frac{\mathbf{x}}{\|\mathbf{x}\|}$ of $\mathbf{b}^{(A)}$ with $\ell_2$-error at most $\sqrt{d}/2$ by obtaining an $\epsilon$-optimal point of $\bar{f}^A$.

Next, we show that we can implement a quantum stochastic gradient oracle of $\bar{f}^A$ using only one query to the oracle $O_A$, given that one can use one query to $O_A$ to implement the following oracle

$$O_{\mathbf{g}}^A |i\rangle |0\rangle \rightarrow |i\rangle |\mathbf{g}_{i,1}\rangle$$

and use one query to $O_{\mathbf{g}}^A$ to implement

$$O_{\hat{\mathbf{g}}}^A |\mathbf{x}\rangle |i\rangle |0\rangle \rightarrow |\mathbf{x}\rangle |i\rangle |\hat{\mathbf{g}}_{i,1}(\mathbf{x})\rangle,$$

where $\hat{\mathbf{g}}_{i,1}(\mathbf{x})$ is defined in (21). Then by applying $O_{\hat{\mathbf{g}}}^A$ to the state

$$\frac{1}{\sqrt{\alpha n}} |\mathbf{x}\rangle \sum_{i \in [\alpha n]} |i\rangle |0\rangle$$

and uncompute the second and the third register, we construct the following stochastic gradient oracle $O_{\hat{\mathbf{g}}}$ of $\bar{f}^A$.

$$O_{\hat{\mathbf{g}}} |\mathbf{x}\rangle \otimes |0\rangle \rightarrow \frac{1}{\sqrt{\alpha n}} |\mathbf{x}\rangle \otimes \sum_{i \in [\alpha n]} |\hat{\mathbf{g}}_{i,1}(\mathbf{x})\rangle.$$

Hence, any quantum algorithm that can find an $\epsilon$-optimum of $\bar{f}^A$ using $T$ queries to $O_{\hat{\mathbf{g}}}$ can be transformed into a quantum algorithm for the $\mathsf{Search}^{\alpha n} \circ \mathsf{Parity}^1$ problem that uses $T$ queries to the oracle $O_A$ defined in (19) and returns an estimate $\widetilde{\mathbf{b}}$ with $\ell_2$-error at most $\sqrt{d}/2$. Then by Lemma 10, for any quantum algorithm making less than

$$\alpha n = \frac{R^2 L^2}{100\epsilon^2}$$

queries to the quantum stochastic gradient oracle $O_{\hat{\mathbf{g}}}$, there exists an $A \in \mathcal{A}_{\alpha n, 1}$ and corresponding $\bar{f}^A$ such that the output of the algorithm is not an $\epsilon$-optimum of $\bar{f}^A$ with probability at least $2/3$. $\quad\square$

