# OpenReview forum: "Quantum speedups for stochastic optimization"
_NeurIPS.cc/2023/Conference — NeurIPS 2023 poster_

### Official Review · Reviewer_PEHT · 2023-07-03

**Soundness:** 3 good
**Presentation:** 3 good
**Contribution:** 3 good
**Rating:** 7
**Confidence:** 3

**Summary:**

The paper considers the problem of stochastic optimisation for both convex and non-convex functions and aims at improving the scaling of the number of queries as a function of the approximation error. The authors introduce a technique based on quantum variance reduction, that builds on top of previous work to estimate the mean of a random variable given access to a quantum sampling oracle.
The paper is well organised and presents three algorithms (Q-AC-SA, Quantum stochastic cutting planes and Q-Spider) where the main contribution is replace access to classical oracle for bounded variance gradient estimator with their quantum oracle.
They show a quadratic speed in some cases at the expense of dependence on the dimensionality d of the optimisation problem.
The authors also provide lower bounds for quantum variance reduction, showing their approach is optimal in this sense.

**Strengths:**

The paper presents novel results that combine the field of quantum variance reduction and that of stochastic optimisation.
The main strength are
- clear organisation of the paper
- relevance of the problem studied.
- mathematically rigorous statements about their claims
- broad applicability of the results (convex to non-convex problems)

**Weaknesses:**

- The scaling of the number of queries of the algorithm introduced now increases with the dimensionality d of the problem.
- The oracle model makes it hard to understand whether practically there are scenarios where this method is favourable that a classical algorithm
- The core routine used, QuantumMeanEstimation, which achieves a better scaling is not explained.
- A few typos (see list below)

Minor:

- Line 58: qubits are superpositions of bit strings, here the authors assume continuous variable. Is the idea to discretise those?
- Line 62: classiic (typo)
- Line 63 and in other places: I found using $\sqrt{dx}$ confusing: I would have written a quantum state $\int dx \sqrt{p(x)} |x\rangle$, otherwise $\langle x|\psi\rangle$ is not a number
- Equation 2: what is $L$?
- $\tilde{O}$ undefined, is it up to polylog?. Why in Theorem 4 we have $\log d$ and not in Lemma 1?
- Line 132: gien (typo)
- Line 133: missing : in definition of f
- Line 163: $\hat{\mu}$ should be $\tilde{\mu}$
- Line 168: there should be their
- Line 183: where is $j\le j_{max}$ used?
- Equation 12: I think that = should be $\le$ and that there is a missing constant $2(1+2\sqrt{2})$ which changes the constant in the bounds
- Line 198: Lemma 3 not defined in the the main text. In general I found the links in the pdf broken.
- You might consider making $g_F(x)$ a function of $y$ explicitly
- Line 217: Missing closing parenthesis in $O(d \log(\epsilon^{-1}))$
- Line 230, 231: x should be bold
- Line 232: $O(d / \log(\epsilon^{-1}))$ should be $O(d \log(\epsilon^{-1}))$
-



**Questions:**

- Could you comment on the implementation of the oracle? What would be the overhead say as a function of d?
- Could you give some intuition around where the quadratic speedup in QuantumMeanEstimation come from?

**Limitations:**

- The fact that the number of queries scales with d limits the application to high dimensional problems, the typical setting of interest in machine learning
- No comment on implementation of the oracle and resources required for implementation on a quantum computer is given

---

> ### Author Rebuttal · Authors · 2023-08-10
>
> Thank you for your positive feedback and detailed suggestions!
>
> Regarding your comment on the $d$ dependence in the number of the queries, we agree that this is good point which has also been pointed out by other reviewers. Please refer to the general response for a detailed discussion on this point.
>
> Regarding your comment on the implementation of the quantum oracle and the utility of our algorithm in practical situations, we agree that you raised a good point which has also been pointed out by other reviewers. Please refer to the general response for a detailed discussion on this point.
>
> Regarding the intuition of the quadratic quantum speedup given by QuantumMeanEstimation, it proceeds by introducing a directional mean function that reduces the multivariate mean estimation problem to the univariate mean estimation problem, which in the bounded norm case can be solved by quantum algorithms with a quadratic speedup using phase estimation. We will include more discussion on this in the final version.
>
> Regarding the typos you pointed out, we sincerely appreciate your careful review, and we will fix them in the final version.

---

> > ### Comment · Reviewer_PEHT · 2023-08-18
> >
> > Thank you for the rebuttal, I do not have further comments.

---

### Official Review · Reviewer_YvE3 · 2023-07-06

**Soundness:** 3 good
**Presentation:** 3 good
**Contribution:** 3 good
**Rating:** 7
**Confidence:** 3

**Summary:**

This paper proposes several quantum algorithms stochastic optimization. First, a technique called "quantum-variance reduction" is introduced and used to develop two new quantum algorithms for convex optimization problems. The "quantum-variance reduction" technique uses a quantum sampling oracle to speedup the task of estimating the mean of a random variable with some desired variance. Additionally, a quantum algorithm for finding critical points of smooth non-convex functions is presented. This algorithm also uses the quantum variance reduction technique and applies it to a classical algorithm for solving the same problem.

**Strengths:**

- Theorems 1 and 2 are good results for the proposed quantum algorithms for convex optimization. It's good that for fixed dimension, the dependence of $\epsilon$ is matched between the algorithm and the quantum lower bound.
- Similarly, Theorem 3 is a good result for the quantum speedup for finding critical points of nonconvex functions.
- The proposed method for quantum variance reduction is interesting, and improves upon previous results by giving an unbiased estimator with better query complexity.
- The application of the quantum variance reduction to several classical algorithms leads to an algorithm with better query complexity than the classical counterparts. This also raises the question of whether this method can be used to improve other algorithms, not necessarily in the domain of stochastic optimization.

**Weaknesses:**

- Since Algorithms 2 and 3 are essentially modified versions of algorithms from existing work, it would be helpful to give a brief description of how each algorithm works in order to make the paper more self-contained.
- Table 1 only compares the proposed quantum algorithms with the classical counterparts. I think other quantum algorithms for solving the same problem should be included as well.
- Typos: "classiic" instead of "classical" in line 62, missing a colon in line 133, "there" in line 168, "wee" in line 290, "provideed" in line 292, "gradient" instead of "gradients" in line 219
- In line 229 (Problem 4), it says $\epsilon \in (0,K)$. Is $K$ the set or a number? This seems to be a typo.


**Questions:**

My main question is related to how the proposed algorithms using quantum variance reduction compare to other quantum algorithms for the same problems. In Line 222, it is said that the result can be obtained using quantum mean estimation from an existing paper, in which case there seems to be no need for quantum variance reduction. However, quantum variance reduction is a major contribution of this work, so naturally I wonder how this method benefits the design of quantum algorithms compared to other quantum algorithms.

**Limitations:**

Algorithm 1 presented in this paper only achieves the optimal scaling with $\epsilon$ when the dimension $d$ is constant. The authors have addressed this as a potential direction for future work.

Since this is a theoretical work on quantum algorithm design, there are no foreseeable negative societal impacts.

---

> ### Author Rebuttal · Authors · 2023-08-10
>
> Thank you for your positive feedback and detailed suggestions!
>
> Regarding your question on the role of our quantum variance reduction technique in stochastic optimization, we use it critically and directly in results Algorithm 2 and Algorithm 3. In particular, Algorithm 2 and Algorithm 3 encounter potential bias issues if instead quantum mean estimation was applied directly, which can impede their convergence. Furthermore, in a broader context, a considerable number of existing gradient-based classical stochastic optimization algorithms require the stochastic gradient to be unbiased. If we intend to extend these algorithms into the quantum domain, the quantum variance reduction technique becomes indispensable in eliminating the bias.
>
> We agree with you that we do not necessarily need to use quantum variance reduction in obtaining Corollary 1, given that stochastic cutting plane method is robust against possible bias. However, it is non-trivial to apply stochastic cutting plane method in our setting and the subsequent analysis is far from straightforward, which is also a technical contribution of our work. We agree with you that you raised a good point and we will clarify more in the final version.
>
> Regarding your comments on the presentation of Algorithm 2 and Algorithm 3, we agree that it is helpful to briefly describe how the corresponding classical algorithms work. We will include this discussion in the final version.
>
> Regarding the comparisons between our quantum algorithms and other quantum algorithms in Table 1, we have diligently searched the literature and, to the best of our knowledge, we have not found any prior quantum algorithms specifically addressing the same problems we are considering. However, if you are aware of any previous quantum algorithm that might be relevant, please kindly let us know; we are very happy to include a comparison in the final version.
>
> Regarding the typos you pointed out, we sincerely appreciate your careful review, and we will fix them in the final version.

---

> > ### Comment · Reviewer_YvE3 · 2023-08-14
> >
> > Thank you for addressing my question. I'm happy with the rebuttal and I wish to keep the score.

---

### Official Review · Reviewer_krht · 2023-07-08

**Soundness:** 1 poor
**Presentation:** 2 fair
**Contribution:** 2 fair
**Rating:** 4
**Confidence:** 3

**Summary:**

This paper tries to give quantum algorithms for solving Lipschitz convex optimization
problems with “quantum stochastic oracle”. Additionally, they claim to have a quantum
algorithm for computing a critical point of a smooth non-convex function at a rate not known
to be achievable classically

**Strengths:**

This paper tries to use the quantum multivariate mean estimator to solve optimization using the quantum stochastic gradient oracle

**Weaknesses:**

Neither the quantum sampling oracle (Definition 1) nor the stochastic quantum gradient oracle (Definition3) is well-defined, and it’s not clear how people can get such a quantum oracle as well. Also, all their algorithms are just applying the quantum multivariate mean
estimation directly to the very well-known classical optimization algorithm. Moreover, the region they can have a quantum speedup is very very limited (assuming all their claims and proofs are correct, which seems unlikely.)


**Questions:**

1. Var[X] In Problem 3 should not be a real value. It should be the covariance matrix
because X is a d-dimensional vector. Also, \hat{\sigma} has nothing to do with Var[X]
and L. So it’s not clear what is the goal of the problem here. Also, if \hat{\sigma} is
smaller than L, this problem is not generally solvable.
2. The query complexity of multivariate mean estimators depends on the trace of the
covariance matrix, while in this paper, they directly assume the trace of the
covariance matrix at most 1, which is very unreasonable.


**Limitations:**

The authors did not adequately address the limitation. I suggest the author first carefully
proofread and check the correctness of the definitions, problems, and the theorems they
cited.

---

> ### Author Rebuttal · Authors · 2023-08-10
>
> Thank you for your detailed comments!
>
> Regarding your question on the notation $\mathrm{Var}[X]$, in our paper it denotes the trace of the  covariance matrix of $X$ and is referred to as the variance of $X$. This colloquial use of the term variance is common in optimization literature and we therefore felt that the term $\mathrm{Var}[X]$ was natural to use. However, we see the confusion this could lead to and will clarify in the final version. Thank you for pointing this out! In a similar vein, $\hat{\sigma}$ symbolizes the square root of the target variance. Regarding the solvability of the problem it is well established that stochastic gradient descent computes $\epsilon$-optimal points of convex functions with only bounds on the $\ell_2$ norm of the minimizer and bounds on the expected squared $\ell_2$-norm of stochastic gradients. If you still have a concern regarding solvability, please let us know.
>
> In response to your inquiry about our assumption concerning the trace distance of the covariance matrix, we wish to clarify that our assumption does not entail a strict bound of $1$. Instead, we assume it is bounded by a specific quantity denoted as $L$, which appears in the overall query complexity.
>
> Regarding your question on the novelty of our algorithm, we would like to clarify that we did not directly apply the quantum multivariate mean estimation directly to classical optimization algorithms. Notably, the quantum mean estimation technique introduces potential bias that accumulates during the iterations of classical algorithms and break their convergence guarantees. As discussed in Section 2, in this work we develop a new algorithm that can obtain unbiased estimates, which is a nontrivial technical contribution that might also be useful in other scenarios.
>
> Regarding your comment on the region where our algorithms have quantum speedups, especially its dependence on the dimension $d$, we agree that this is good point which has also been pointed out by other reviewers. Please refer to the general response for a detailed discussion on this point.
>
> Regarding your comment on the implementation of the quantum oracle and the utility of our algorithm in practical situations, we agree that you raised a good point which has also been pointed out by other reviewers. Please refer to the general response for a detailed discussion on this point.
>
> Regarding your comment on the correctness of our proof, we wish to assure you that we have conducted a thorough proofreading process. In this process we encountered some corner cases, e.g., how we handle expected error versus high probability bounds, and we plan to address all in the final version. Nevertheless, if you have any concern regarding any specific part of the paper, please kindly let us know, We are more than willing to perform an additional check to ensure the correctness of our results.

---

> > ### Comment · Reviewer_krht · 2023-08-15
> >
> > I thank the authors for their response. I think it answers a few of my concerns, and I will raise my rating a little bit. I still feel the main technical contributions in this paper aren't strong (clearly they needed to modify well-known quantum subroutines to make them work for their application).

---

### Official Review · Reviewer_ewVR · 2023-07-11

**Soundness:** 3 good
**Presentation:** 3 good
**Contribution:** 3 good
**Rating:** 5
**Confidence:** 2

**Summary:**

The authors focus on the optimization problem when quantum access is given to a stochastic oracle. They show that quantum speedups are possible for dimension-dependent algorithms and develop a quantum algorithm for computing the critical point of a smooth non-convex function along with a general quantum-variance reduction technique.

**Strengths:**

- The authors present multiple different quantum-based approaches in different settings along with the theory. Quantum variance reduction for stochastic nonconvex optimization is a novel contribution as far as I know.

- The authors present theoretical advancements obtained by their proposed algorithms in Table 1. It looks technically correct but I might have missed some points.

**Weaknesses:**

The main advancements in the theoretical bounds are obtained by adding the additional dimension restriction, it looks technically correct however in practice I am not convinced whether it would be possible to maintain the desired dimensions that benefit the presented algorithm.

Writing related:

- In writing, there is a lot of repetition; abstract, lines 42-48, lines 102-105 are almost the same.

- The results are shared in Table 1, before specific definitions of the parameters are given (epsilon and d are defined after), which is difficult to follow.

Minor :

- Typo in line 62 "classiic" -> "classic", 290 "wee"->"we", 292 "provideed"-> "provided"

**Questions:**

- The success of the quantum oracle is shown by the number of queries required to solve the problem. However there are some further aspects, like how much time each query takes, or also maybe the permutations of the queries (in random sampling since each item has equal probability showing the convergence is simple but how is it in quantum based samplings?), which subset of the queries to take (~active learning). Can the authors comment on these aspects too?

- Is the contribution purely theoretic or are there any empirical results that the authors can show?

**Limitations:**

This paper does not have a direct potential negative societal impact.

---

> ### Author Rebuttal · Authors · 2023-08-10
>
> Thank you for your positive feedback and detailed suggestions!
>
> Regarding your comment on the definition of our quantum oracle and the utility of our algorithm in practical situations, we agree that you raised a good point which has also been pointed out by other reviewers. Please refer to the general response for a detailed discussion on this point.
>
> Regarding your comments concerning the detailed implementation of the quantum oracle in various scenarios, such as random sampling or active learning, we agree that that these are intriguing questions for future work. and we may expand on this in the final version.
>
> Regarding the typos and repetitions you pointed out, we sincerely appreciate your careful review, and we will fix them in the final version. Regarding your comments on the order between Table 1 and the definitions of parameters, we thank you for pointing this out and we will fix this in the final version.

---

> > ### Comment · Reviewer_ewVR · 2023-08-14
> >
> > I read the rebuttal response. Thank you for addressing my points.
> >
> > No change in my current evaluation.
> >
> > Expanding the final version mentioning the further work and addressing the minor comments would make the paper ready.

---

### Official Review · Reviewer_Zhqz · 2023-07-29

**Soundness:** 3 good
**Presentation:** 3 good
**Contribution:** 3 good
**Rating:** 7
**Confidence:** 4

**Summary:**

The paper presents quantum optimization algorithms for convex and non-convex optimization. The algorithms access a certain quantum stochastic oracle defined by the objective function f, and the presented complexity is lower than the best possible results in its classical counterparts.

**Strengths:**

The formulation is interesting. To the reviewer’s knowledge, this work is one of few that aims to find the fundamental limits of optimization algorithms in a quantum setting.

**Weaknesses:**

The formulation requires one particular quantum oracle (equation 3), where given any input state x, the “phase” of the outcomes is known, or particularly the oracle returns a pure state. This is strictly stronger than the classical formulation, especially since a classical algorithm would apply when the observation follows equation (1) and the garbage states are orthogonal (i.e., the density matrix being diagonal instead of being rank 1). So it is not immediately clear to the readers whether the reduced complexity is achieved by using quantum algorithms or just because we are dealing with an easier problem. If it is the second case, or if the proposed algorithm will not apply when the quantum phase information is not known. This distinction should be clearly mentioned in the abstract.

**Questions:**

Please see the comment on weakness.

**Limitations:**

It is possible that the limitations of the proposed approach should be emphasized. Please see the comment on weakness.

---

> ### Author Rebuttal · Authors · 2023-08-10
>
> Thank you for your positive feedback and detailed suggestions!
>
> Regarding your comment on the definition of our quantum oracle and the utility of our algorithm in practical situations, we agree that you raised a good point which has also been pointed out by other reviewers. Please refer to the general response for a detailed discussion.
>
> Regarding your question on the "phase'' of the outcome, it is in general unknown in our setting, given that the garbage states can be arbitrary and contain arbitrary phases. We acknowledge our uncertainty regarding the context you may be referring to. If there is an alternative setting that you have in mind, we kindly request that you provide further clarification or details. We are happy to add more discussion in the final version.
>
> In regards to the question of "quantum algorithms" versus an "easier problem," note that given a standard (classical) stochastic gradient oracle, the rate of $O(\epsilon^{-2})$ for convex optimization is unimprovable. Consequently, in order to achieve improved complexities, it is essential that we have a more powerful oracle for accessing the function. Note that our focus in this paper is on the query complexity of optimization given this oracle, rather than the additional computation involved in using the oracle. We will be sure to check that our final abstract is clear on the point that we are not working with the standard classical gradient oracle.

---

> > ### Comment · Reviewer_Zhqz · 2023-08-20
> >
> > Thank you for the clarification. Assuming the abstract will be updated as discussed if accepted, I've raised my score by 1.

---

### Author Rebuttal · Authors · 2023-08-10

Thank you to the reviewers for their thoughtful consideration of our paper. We appreciate your positive recognition of our contributions and helpful suggestions for improvement. Here we address some common questions that arose in multiple reviews.

We noticed that the reviewers had some questions in common regarding our submission; here we provide answer these question. (More Specific questions of individual reviewers are provided in individual responses.)

$\mathbf{Practicality. }$
Regarding the implementation of our quantum oracle and the utility of our algorithm in practical situations, we note that our quantum oracles are defined as direct, natural extensions of the corresponding classical oracles. Connsidering such quantum generalizations of classical oracles i standard in the literature, see e.g., [1, 2]. Furthermore, there are standard techniques for implementing such quantum analogs of classical oracles (in theory for now given the current state-of-the-art in implementing quantum algorithms in practice). In particular, if there is a classical circuit for the classical oracle, there is a standard technique to use a quantum circuit of the same size to implement the corresponding quantum oracle. Hence, we believe our quantum algorithms have the potential to surpass blackbox classical algorithms in low dimensional settings where the oracle is given as an explicit circuit. We will add more discussion on this point in the final version.

Nevertheless, given the current state-of-the-art on quantum computation in practice,  this work is purely theoretical for now. We agree that until more practical quantum computational resources are widely available, the practical utility of our advances is indeed unclear. We will include more discussion on this in the final version, and will add this as a limitation.

$\mathbf{Dimension\ dependence. }$
Regarding the parameter region where our algorithms have quantum speedups and their dependence on the dimension $d$, note that our algorithms apply for all $\epsilon$ and $d$, i.e., there is no dimension restriction on whether they obtain the claimed bounds. However, our claimed bounds only improve upon state-of-the-art classical algorithms when $\epsilon$ is sufficiently small relative to the dimension. Below we provide several additional points towards addressing this concern (all of which will be addressed in the final version):

(1) As mentioned, In the classical setting, prior research [3] has demonstrated that when optimizing an 1-Lipschitz convex function, SGD with query complexity $O(\epsilon^{-2})$ remains optimal, even in the one-dimensional case, and this $\epsilon$-dependence remains unaffected by the inclusion of additional factors, such as dimension $d$. In the quantum setting however, we addressed an interesting question about when quantum speedups are possible and discovered theoretical phenomenon that there exists a tradeoff between $\epsilon$ dependence and $d$ dependence. From our lower bound we know that some dimension dependence is inherent in obtaining an improvement.

(2) Classically, the question of utility dimension dependent is a long line of work. In particular, there are parallel and private stochastic convex settings where the dimension dependence is discussed, see e.g., [4, 5], and there exists works on critical point computation in low $d$ settings, see e.g., [6].

(3) There is the possibility of applying our algorithm as a subroutine in settings where the dimension is lower. For instance, consider its application to the approximate optimal finding problem (Problem 5), wherein we utilize our algorithm repeatedly within a one-dimensional setting.

$\mathbf{References. }$

[1]. Shouvanik Chakrabarti, Andrew M. Childs, Tongyang Li, and Xiaodi Wu. "Quantum algorithms and lower bounds for convex optimization."

[2]. Chenyi Zhang, Jiaqi Leng, and Tongyang Li. "Quantum algorithms for escaping from saddle points."

[3]. John C. Duchi. "Introductory lectures on stochastic optimization."

[4]. Yair Carmon, Arun Jambulapati, Yujia Jin, Yin Tat Lee, Daogao Liu, Aaron Sidford, and Kevin Tian. "Resqueing parallel and private stochastic convex optimization."

[5]. Sébastien Bubeck, Qijia Jiang, Yin Tat Lee, Yuanzhi Li, and Aaron Sidford. "Complexity of highly parallel non-smooth convex optimization."

[6]. Sinho Chewi, Sébastien Bubeck, and Adil Salim. "On the complexity of finding stationary points of smooth functions in one dimension."

---

### Decision · Program_Chairs · 2023-09-21

**Decision:**

Accept (poster)

**Comment:**

The paper presents quantum algorithms for stochastic optimization problems, both convex and non-convex, using a technique called “quantum variance reduction.” The reviewers have generally provided positive feedback on the paper, and the authors have responded adequately to the reviewers' comments and concerns.

The paper presents novel results that combine the fields of quantum algorithms and stochastic optimization and offer promising potential for quadratic speedup in specific scenarios, which is a notable contribution. Further strengths of the paper, as highlighted by the reviewers, include its clear organization, the relevance of the problem studied, and its exceptional mathematical rigor.

However, there are some limitations, particularly the scaling of the number of queries with the dimensionality of the problem, which may limit the application to high-dimensional problems – the relevant scenario in machine learning. Additionally, the paper does not provide insights into the practical implementation of the quantum oracle and the resources required for quantum computation; the reviewers have raised questions about the implementation of the oracle, the intuition behind the quadratic speedup achieved in QuantumMeanEstimation, and comparisons with other quantum algorithms for the same problems.

Overall, the paper is technically sound and presents valuable contributions to the field of quantum optimization. The authors should carefully address the major concerns raised by the reviewers and provide additional clarity in the final version.